# GENERALIZABLE CROSS-MODALITY DISTILLATION WITH CONTRASTIVE LEARNING

## ABSTRACT

Cross-modality distillation arises as an important topic for data modalities containing limited knowledge such as depth maps and high-quality sketches. Such techniques are of great importance, especially for memory and privacy-restricted scenarios where labeled training data is generally unavailable. To solve the problem, existing label-free methods leverage a few pairwise unlabeled data to distill the knowledge by aligning features or statistics between the source and target modalities. For instance, one typically aims to minimize the L2 distance between the learned features of pairs of samples in the source (e.g. image) and the target (e.g. sketch) modalities. However, these approaches only consider the positive correspondence in paired samples, which is typically limited in quantity, while overlooking the potential information within the negative relationship present in the unpaired data, which is more abundant in cross-modality datasets. To exploit such a negative relationship which plays a vital role in learning discriminative feature representation, we propose a novel framework called generalizable cross-modality contrastive distillation (CMCD), built upon contrastive learning that leverages both positive and negative correspondence, towards a better distillation of generalizable features. Extensive experimental results show that our algorithm outperforms existing algorithms consistently by a margin of 2-3% across diverse modalities and tasks, covering modalities of image, sketch, depth map, and audio and tasks of recognition and segmentation. Our convergence analysis reveals that the distance between source and target modalities significantly impacts the test error on downstream tasks within the target modality which is also validated by the empirical results.

## 1 INTRODUCTION

Cross-modality distillation is a significant topic in machine learning and deep learning, which distills the 'rich' knowledge in one modality to improve the modality of 'limited' knowledge (Patel et al., 2015; Gupta et al., 2016; Thoker & Gall, 2019; Ahmed et al., 2022). However, most existing cross-modality methods (Patel et al., 2015; Saito et al., 2020; Liang et al., 2020; Gupta et al., 2016; Thoker & Gall, 2019; Wang et al., 2022) require labeled data in the source modality, which may not be available in real scenarios due to memory or privacy constraints. To address this challenge, some recent approaches (Zhao et al., 2020; Ahmed et al., 2022) propose to distill knowledge without label information and align features or statistics between the source and target modalities using a few paired data instead. For instance, given images as the source modality and sketches as the target, a common method is to minimize the L2 distance between the learned features extracted from the images and sketches. Unfortunately, such a method only leverages positive correspondences, whilst ignoring negative relationships among the

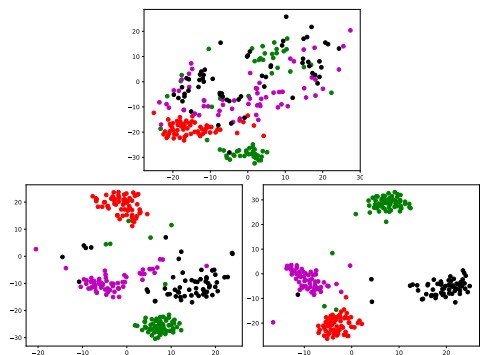

Figure 1: t-SNE embeddings of feature from direct SimCLR (Chen et al., 2020b) (top), CMKD (Zhao et al., 2020)(bottom-left), and our CMCD (bottom-right).

data. Specifically, for each sketch, there is only one positive image that matches it, while many other images are negative choices. Previous studies (Schroff et al., 2015; He et al., 2018), conducted with paired data, have convincingly demonstrated the pivotal role of negative relationships in the acquisition of discriminative feature representations. Furthermore, the process of amassing such pairwise data is both demanding in terms of time and labor. For example, creating a sketch for a given image can be a time-consuming task, potentially resulting in an incomplete representation of the data distribution, primarily focusing on positive relationships. Consequently, relying solely on positive correspondences for cross-modality distillation may prove inadequate in attaining the acquisition of versatile and discriminative features within the target modality. Thus, it is shown in Figure 1 that the feature distributions learned by these algorithms lack a distinct separation.

To overcome this limitation, we propose a versatile framework called generalizable Cross-Modality Contrastive Distillation (CMCD) based on contrastive learning in this paper. The CMCD framework tries to investigate both the positive and negative relationships in a contrastive distillation way that enables the efficient transfer of generalizable features from the source modality to the target modality. Concretely, we introduce two types of loss functions based on contrastive learning for conducting cross-modality contrastive distillation. One of them is designed upon knowledge distillation techniques and the other is inspired by multi-modality pretraining works such as CLIP(Radford et al., 2021). We demonstrate that our algorithms can outperform other cross-modality transfer algorithms across various modalities (e.g., images, sketches, depth maps, videos, and audio) and tasks such as recognition and segmentation.

In addition to empirical results, we also establish theoretical analysis for our framework. One well-known theoretical result in transfer learning (Ben-David et al., 2010) states that when applying domain adaptation methods, the test error of the target modality can be bounded by the test error of the source modality and the $\mathcal{H}\Delta\mathcal{H}$ divergence between the source and target modalities. However, this result is not directly applicable to contrastive learning-based algorithms. Moreover, existing theoretical research on contrastive learning (Tosh et al., 2021; Saunshi et al., 2022; Ge et al., 2023) primarily focuses on single modality scenarios. Therefore, to analyze our algorithm, we develop novel theories that merge cross-modality learning and contrastive learning. Through our analysis, we give the generalization bound of each step in our algorithm and introduce a final theorem that illustrates the final test error of the downstream task in the target modality will be bounded by the total variation distance between the source and target modalities. In essence, the theorem reveals that as the total variation distance between the modalities decreases, our algorithm has a higher likelihood of obtaining generalizable features in the target modality. This theoretical insight is further validated by our experimental results. In summary, our contributions are as follows:

- A novel generalizable cross-modality contrastive distillation (CMCD) framework is proposed to utilize both positive and negative relationships in paired data and effectively distill generalizable representation from a source modality to a specific target modality.
- We conducted extensive experiments that provide compelling evidence for the efficacy of our algorithm across diverse modalities and a range of downstream tasks.
- We perform a theoretical analysis of our algorithm, elucidating the algorithm's convergence bounds. Our findings underscore a direct correlation between the algorithm's ultimate performance and the total variation distance between the source and target modalities that is further validated by our empirical results.

## 2 RELATED WORKS

**Cross-modality transfer/distillation**. Cross-modality distillation, which can be viewed as a form of transfer learning, has been a subject of study for a considerable period (Gupta et al., 2016; Zhao et al., 2020; Xue et al., 2021; 2022). In the realm of deep learning, Gupta et al. (2016) propose a distillation method that relies on feature alignment to transfer supervision from RGB images to depth images. Building upon this work, CMKG (Zhao et al., 2020) encoded teacher network knowledge into priors through meta-learning. SOCKET (Ahmed et al., 2022) proposes to minimize the distance between the statistics of features instead of the original feature activation maps. Rather than focusing on an unimodal student, MKE (Xue et al., 2021) leveraged multimodal pairs to enhance the student network. Wang et al. (2023) introduced a generic pretraining framework that incorporates knowledge from multiple modalities using cross-modality contrastive learning. However, their

method necessitates a substantial amount of paired data. Another line of research explores dataset distillation (Lei & Tao, 2023), which aims to select or generate the most informative samples for feature embedding in the target modality. In our research, we aim at transferring the knowledge from a source modality to a target modality which is both unsupervised and uses a small number of paired data to distill the knowledge.

**Contrastive learning**. Contrastive learning has emerged as a popular and effective technique in self-supervised and unsupervised tasks (Oord et al., 2018; Wu et al., 2018; Chen et al., 2020b; He et al., 2020). At its core, contrastive learning trains models to capture discriminative features among data instances, typically achieved through data augmentation (Chen et al., 2020b) and the InfoNCE loss (Oord et al., 2018). Methods such as SimCLR/SimCLR v2 (Chen et al., 2020b;c) propose simple frameworks with various data augmentations for contrastive learning. Momentum Contrast (He et al., 2020) introduced a momentum encoder to enhance negative pairs' features from a memory bank. Additionally, BYOL (Grill et al., 2020) maintains online and target networks, updating the target network with a slow-moving average of the online network. More recently, foundational models like CLIP (Radford et al., 2021) have connected images and language through contrastive learning. However, contrastive learning approaches are primarily designed for self-supervised pre-training on large-scale datasets. In this paper, we explore how contrastive learning can serve as a powerful method for transferring knowledge from a major modality, such as images, to a minor modality, such as high-quality sketches.

**Theoretical results**. We discuss the theoretical results of Contrastive Learning and Transfer Learning. Compared to the experimental research, there is a limited theoretical analysis of cross-modality knowledge distillation. MFH (Xue et al., 2022) proposes a hypothesis that emphasizes the importance of overlapped information between two domains for successful distillation, supported by a simple proof in the linear case. To contextualize our research, we also review theoretical results from the fields of contrastive learning and transfer learning. Wen & Li (2021) prove that contrastive learning with ReLU activation can learn desirable sparse features when appropriate augmentations are used. In Saunshi et al. (2022), they elaborate on the importance of inductive biases in the analysis of contrastive learning. The most related analysis to our research is Ge et al. (2023), they give a complexity-based convergence bound of contrastive learning which indicates the benefits of the pretraining but is not concerned about the cross-modality contrastive case. For transfer learning, a basic theoretical result from Ben-David et al. (2010) states that target error can be bounded by the source error and the $\mathcal{H}\Delta\mathcal{H}$ divergence between source and target distributions under a supervised domain adaptation setting. Tripuraneni et al. (2020) formulate the target error by using the task-representation difference which can be used to measure the task diversity.

## 3 METHODOLOGY

**Notations.** We define the source modality and target modality as $\mathcal{A}$ and $\mathcal{B}$. We denote by $x_\mathcal{A} \in \mathcal{X}_\mathcal{A}, x_\mathcal{B} \in \mathcal{X}_\mathcal{B}$ the input data of two modalities respectively. In our setting, we do not require the supervised labels for both the source modality $\mathcal{A}$ and target modality $\mathcal{B}$ to perform cross-modality distillation. Subsequently, when performing a downstream task in modality $\mathcal{B}$, a few labels $y_\mathcal{B} \in \mathcal{Y}_\mathcal{B}$ are needed for fine-tuning the model. Our goal is to get an efficient model of the downstream task in target modality $\mathcal{B}$, i.e., a model can predict $y_\mathcal{B}$ from $x_\mathcal{B}$. We assume that $y$ is connected to $x$ through a latent variable/feature $z$ which means that given $z$ the value of $y$ is independent of $x$. In order to construct the theoretical analysis for our algorithm, we follow the setting of Ge et al. (2023) and introduce the side information $s \in \mathcal{S}$ which can be accessed from $x \in \mathcal{X}$. For example, in contrastive learning, given $(x, x') \in \mathcal{X}^2$, $s := \mathbb{1}(x = x')$ where the side information indicates whether the pair should be considered as positive or negative. To measure the distance between two distributions, we use total variation distance $d_{TV}(\mathbb{P}, \mathbb{Q}) = \int |p(x) - q(x)|dx$.

We use two kinds of models to construct distributions of $(x, z, y)$, the latent variable model $\phi$ modeling the relationship between $(x, z)$ and the prediction model $\psi$ modeling the relationship between $(z, y)$. Concretely, we assume that there exist oracle models $\phi_\mathcal{A}^*, \phi_\mathcal{B}^*, \psi_\mathcal{B}^*$ for both source and target modalities $\mathcal{A}, \mathcal{B}$, indicating $z_\mathcal{A} = \phi_\mathcal{A}^*(x_\mathcal{A}), z_\mathcal{B} = \phi_\mathcal{B}^*(x_\mathcal{B}), y_\mathcal{B} = \psi_\mathcal{B}^*(z_\mathcal{B})$.

### 3.1 FRAMEWORK OF GENERALIZABLE CROSS-MODALITY CONTRASTIVE DISTILLATION

**Step 1.** Firstly, given the source modality $\mathcal{A}$ with massive unlabeled data $S_{\mathcal{A}} = \{x_i^{\mathcal{A}}\}_{n_{\mathcal{A}}}$, we can use typical contrastive learning such as SimCLR to learn the latent feature representation $\hat{\phi}_{\mathcal{A}}$. Specifically, we use the InfoNCE (Wu et al., 2018) loss to train the model $\phi_{\mathcal{A}}$:

$$\mathcal{L}_{\text{InfoNCE}} = -\sum_{i,j} \log \frac{\exp(z_i^{\mathcal{A}} \cdot z_j^{\mathcal{A}}/\tau)}{\sum_t \exp(z_t^{\mathcal{A}} \cdot z_j^{\mathcal{A}}/\tau)} \tag{1}$$

where $\tau$ is the temperature hyper-parameter, and $z_i^{\mathcal{A}} = \phi_{\mathcal{A}}(x_i^{\mathcal{A}})$ is a projected feature by the model $\phi_{\mathcal{A}}$. Numerous studies (Chen et al., 2020b; He et al., 2020; Grill et al., 2020) have demonstrated that self-supervised models can serve as effective feature extractors for various downstream tasks.

**Step 2.** Secondly, we leverage the pair data of source and target modalities $S_{\mathcal{AB}} = \{(x_i^{\mathcal{A}}, x_i^{\mathcal{B}})\}_m$ to distill the information from the source modality to the target modality. In this cross-modality distillation step, we propose two types of cross-modality losses. The first one is based on knowledge distillation and is referred to as the cross-modality distillation (CMD) loss:

$$\mathcal{L}_{\text{CMD}} = -\sum_{i,j} \frac{\exp(z_i^{\mathcal{A}} \cdot z_j^{\mathcal{A}}/\tau)}{\sum_t \exp(z_t^{\mathcal{A}} \cdot z_j^{\mathcal{A}}/\tau)} \log \frac{\exp(z_i^{\mathcal{B}} \cdot z_j^{\mathcal{B}}/\tau)}{\sum_t \exp(z_t^{\mathcal{B}} \cdot z_j^{\mathcal{B}}/\tau)} \tag{2}$$

where $z_i^{\mathcal{A}} = \hat{\phi}_{\mathcal{A}}(x_i^{\mathcal{A}})$ and $z_i^{\mathcal{B}} = \phi_{\mathcal{B}}(x_i^{\mathcal{B}})$ represent the learned feature in source modality and the project feature from target modality to be optimized. This distillation-type loss is proposed for self-supervised distillation first (Fang et al., 2021) which aims at distilling the information from a large model (e.g., ResNet101) to a small model (e.g., ResNet18) without any supervision. The second loss is inspired by CLIP (Radford et al., 2021) and is called cross-modality contrastive (CMC) loss:

$$\mathcal{L}_{\text{CMC}} = -\sum_i \left( \log \frac{\exp(z_i^{\mathcal{A}} \cdot z_i^{\mathcal{B}}/\tau)}{\sum_t \exp(z_t^{\mathcal{A}} \cdot z_i^{\mathcal{B}}/\tau)} + \log \frac{\exp(z_i^{\mathcal{B}} \cdot z_i^{\mathcal{A}}/\tau)}{\sum_t \exp(z_t^{\mathcal{B}} \cdot z_i^{\mathcal{A}}/\tau)} \right) \tag{3}$$

This loss was originally used for multi-modality pretraining which needs a lot of paired data, while in our algorithm we utilize it to transfer latent features from the source modality to the target modality which needs much less paired data than pretraining. As will be demonstrated, both CMD and CMC losses work well for cross-modality knowledge distillation theoretically and experimentally.

**Step 3.** After distillation, we can use the learned feature representation $\hat{\phi}_{\mathcal{B}}$ in the target modality to solve downstream tasks (e.g., classification, semantic segmentation) with some simple fine-tuning, i.e., training a one-layer classifier based on the features. For instance, we can utilize the model $\hat{\phi}_{\mathcal{B}}$ and a small number of labels $y_i^{\mathcal{B}}$ to train an MLP for a classification task. We formulate this step using cross entropy loss in Step 3 in Algorithm 1. However, this task can be replaced by any other downstream tasks such as semantic segmentation or detection with different labels and loss functions. The overall algorithm flow is summarized in Algorithm 1 below.

### 3.2 THEORETICAL ANALYSIS

In this section, we prove that the test error of the downstream task in target modality $\mathcal{B}$ can be bounded in probability by the total variation of the latent feature distribution between source and target modality, i.e., $d_{TV}(\mathbb{P}_{\phi_{\mathcal{B}}^*}, \mathbb{P}_{\phi_{\mathcal{A}}^*})$ and the Rademacher complexities related to $\Phi_{\mathcal{B}}$ and $\Psi_{\mathcal{B}}$, respectively. We mainly analyze the CMD loss in the main text and the discussion for CMC loss can be found in Appendix B. To begin with, we introduce an assumption that builds a relationship between the contrastive loss and the downstream loss.

**Assumption 3.1.** *($\kappa^{-1}$-informative condition.) We assume that the model class $\Phi$ is $\kappa^{-1}$-informative with respect to the true models $\phi^*, \psi^*$ if for any $\phi \in \Phi$, and $x \in \mathcal{X}$, such that*

$$\mathcal{L}_{CE}(\psi^* \circ \phi(x), y) \leq \kappa \mathbb{E}_{x'}[\mathcal{L}_{CMD}(\phi, \phi^*, (x, x'), s)] \tag{7}$$

*where $\mathcal{L}_{CM}(\phi, \phi^*, (x, x'), s) := \mathcal{L}_{CM}(\phi(x), \phi(x'), \phi^*(x), \phi^*(x'))$ for notation simplicity.*

---

**Algorithm 1:** Generalizable Cross-Modality Contrastive Distillation

---

**Data:** $S_\mathcal{A} = \{x_i^\mathcal{A}\}_{n_\mathcal{A}}, S_\mathcal{B} = \{(x_i^\mathcal{B}, y_i^\mathcal{B})\}_{n_\mathcal{B}}, S_{\mathcal{AB}} = \{(x_i^\mathcal{A}, x_i^\mathcal{B})\}_m$

**Result:** $\hat{\phi}_\mathcal{B}, \hat{\psi}_\mathcal{B}, \hat{\phi}_\mathcal{A}$

**Step 1**: Contrastive learning of source modality $\mathcal{A}$

$$\hat{\phi}_\mathcal{A} = \underset{\phi \in \Phi_\mathcal{A}}{\arg\min} \sum_{i,j=1}^{n_\mathcal{A}} \mathcal{L}_{\text{InfoNCE}}(\phi(x_i), \phi(x_j), s_{ij}) \tag{4}$$

**Step 2**: Distillation: contrastive distillation of $\mathcal{A}, \mathcal{B}$ to an error of $\epsilon_{\mathcal{AB}}$

$$\hat{\phi}_\mathcal{B} = \underset{\phi \in \Phi_\mathcal{B}}{\arg\min} \sum_{i,j=1}^{n} \mathcal{L}_{\text{CM}}(\phi(x_i), \phi(x_j), \hat{\phi}_\mathcal{A}(x_i), \hat{\phi}_\mathcal{A}(x_j)), \quad (\mathcal{L}_{\text{CM}} = \mathcal{L}_{\text{CMD}} \text{ or } \mathcal{L}_{\text{CMC}}) \tag{5}$$

**Step 3**: fine-tune on the target modality $\mathcal{B}$ to an error of $\epsilon_\mathcal{B}$

$$\hat{\psi}_\mathcal{B} = \underset{\psi \in \Psi_\mathcal{B}}{\arg\min} \sum_{i=1}^{n_\mathcal{B}} \mathcal{L}_{\text{CE}}(\psi \circ \hat{\phi}_\mathcal{B}(x_i), y_i) \tag{6}$$

---

It is introduced in Ge et al. (2023) to guarantee the feature extraction model $\phi$ and the side information $s$ contains a certain level of information that can reveal the relationship between $x$ and $z$. In other words, this assumption implies that the model obtained with contrastive learning performs reasonably well on downstream tasks. Here we assume that it holds true for both source and target modalities.

In order to derive the whole generalization bound of our algorithm, we start with the bound of the contrastive learning step, i.e., Step 1 of Algorithm 1.

**Lemma 3.1.** *Let $\hat{\phi}_\mathcal{A}$ the minimizer of equation 4. Then, with probability at least $1 - \delta$, we have,*

$$d_{TV}(\mathbb{P}_{\hat{\phi}_\mathcal{A}}(\boldsymbol{x}, s), \mathbb{P}_{\phi_\mathcal{A}^*}(\boldsymbol{x}, s)) \le 3\sqrt{\frac{1}{n_\mathcal{A}^2} \log \frac{N_{[]}(P_{\mathcal{X}_\mathcal{A} \times \mathcal{S}}(\Phi_\mathcal{A}), \frac{1}{n_\mathcal{A}^2})}{\delta}} \tag{8}$$

*where $\mathcal{P}_{\mathcal{X}_\mathcal{A} \times \mathcal{S}}(\Phi_\mathcal{A}) = \{\mathbb{P}_{\phi_\mathcal{A}}(\boldsymbol{x}, s) | \phi_\mathcal{A} \in \Phi_\mathcal{A}\}$, $\boldsymbol{x} = (x_i, x_j)$, $s$ indicates whether $x_i, x_j$ is the paired data, and $N_{[]}(\cdot, \cdot)$ denotes the bracket number. Here the density function of the distribution $\mathbb{P}_\phi(\boldsymbol{x}, s)$ is defined by,*

$$p_\phi(\boldsymbol{x}, s) = \frac{\exp(z_i \cdot z_j / \tau)}{\sum_t \exp(z_t \cdot z_j / \tau)}; z_i = \phi(x_i) \tag{9}$$

which can be seen as a Gibbs distribution of modeling the paired data in a contrastive learning view. This lemma states that with contrastive learning the total variation distance between the best feature representation $\phi_\mathcal{A}^*$ and the learned feature representation $\hat{\phi}_\mathcal{A}$ can be bounded by the bracket number of the possible distribution space $\mathcal{P}_{\mathcal{X}_\mathcal{A} \times \mathcal{S}}(\Phi_\mathcal{A})$. The proof of this lemma is based on a reformulation of the contrastive learning task into a maximum likelihood estimation task. Detailed proof of this lemma can be found in Appendix A.1

Now we bound the test error of the CMD loss learned from the contrastive distillation step, i.e., Step 2 of Algorithm 1.

**Theorem 3.2.** *Let $\hat{\phi}_\mathcal{B}$ the minimizer of equation 5. Assume that $\sup_{\phi_\mathcal{B} \in \Phi_\mathcal{B}, \boldsymbol{x}_{ij}} \langle \phi_\mathcal{B}(\boldsymbol{x}_i), \phi_\mathcal{B}(\boldsymbol{x}_j) \rangle \le B$. Then given $\hat{\phi}_\mathcal{A}$, with probability at least $1 - \delta$, we have*

$$\mathbb{E}[\mathcal{L}(\hat{\phi}_\mathcal{A}, \hat{\phi}_\mathcal{B}, \boldsymbol{x}, s)] - \frac{1}{m^2} \sum_{i,j=1}^{m} \mathcal{L}(\hat{\phi}_\mathcal{A}, \hat{\phi}_\mathcal{B}, \boldsymbol{x}_{ij}, s_{ij}) \le 2R_{m^2}(\mathcal{L} \circ \Phi_\mathcal{B}) + B\sqrt{\frac{2\ln(1/\delta)}{m^2}} \tag{10}$$

*where $\mathcal{L}(\hat{\phi}_\mathcal{B}, \hat{\phi}_\mathcal{A}, \boldsymbol{x}_{ij}, s_{ij}) := \mathcal{L}_{CMD}(\phi_\mathcal{B}(x_i), \phi_\mathcal{B}(x_j), \phi_\mathcal{A}(x_i), \phi_\mathcal{A}(x_j))$ for notation simplicity.*

This theorem describes a similar result of the oracle inequality of ERM where the difference between the test and empirical loss can be bounded by the Rademacher complexity $R_{m^2}(\mathcal{L} \circ \Phi_\mathcal{B})$ and an $\mathcal{O}(m^{-1})$ term.

*Proof Sketch*: This bound comes from Mcdiarmid's inequality (Zhang, 2023) and the definition of the Rademacher complexity. By constructing the function $f(\boldsymbol{x}_{11}, \ldots, \boldsymbol{x}_{mm}) = \sup_{\phi_\mathcal{B} \in \Phi_\mathcal{B}}[\mathbb{E}[\mathcal{L}(\hat{\phi}_\mathcal{A}, \hat{\phi}_\mathcal{B}, \boldsymbol{x}, s)] - \frac{1}{m^2}\sum_{i,j=1}^{m}\mathcal{L}(\hat{\phi}_\mathcal{A}, \hat{\phi}_\mathcal{B}, \boldsymbol{x}_{ij}, s_{ij})]$, we can prove that $f$ satisfy the bounded difference property and apply Mcdiarmid's inequality on $f$ to prove the final results. Detailed proof can be found in the Appendix A.2.

Finally, we prove that the test error of the downstream task in target modality $\mathcal{B}$ can be bounded in probability by the total variation of the latent feature distribution between source and target modality, i.e., $d_{TV}(\mathbb{P}_{\phi_\mathcal{B}^*}, \mathbb{P}_{\phi_\mathcal{A}^*})$ and the Rademacher complexities related to $\Phi_\mathcal{B}, \Psi_\mathcal{B}$ respectively.

**Theorem 3.3.** *Let $\hat{\phi}_\mathcal{B}$ and $\hat{\psi}_\mathcal{B}$ be the outputs of Algorithm 1. Suppose the loss function $\mathcal{L}$ is $L$-bounded, $\sup_{\phi_\mathcal{B} \in \Phi_\mathcal{B}, \boldsymbol{x}_{ij}} \langle \phi_\mathcal{B}(x_i), \phi_\mathcal{B}(x_j) \rangle \leq B$ and the model follows $\kappa^{-1}$-informative. Then with probability at least $1 - \delta$, the test risk of $\hat{\phi}_\mathcal{B}$ and $\hat{\psi}_\mathcal{B}$ is bounded by*

$$\mathbb{E}[\mathcal{L}(\hat{\psi}_\mathcal{B} \circ \hat{\phi}_\mathcal{B}(x), y)] \leq \kappa B \cdot d_{TV}(\mathbb{P}_{\phi_\mathcal{B}^*}, \mathbb{P}_{\phi_\mathcal{A}^*}) + \kappa \epsilon_{\mathcal{A}\mathcal{B}} + \epsilon_\mathcal{B} \tag{11}$$

$$+ 2\kappa R_{m^2}(\mathcal{L}_{CMD} \circ \Phi_\mathcal{B}) + 2R_{n_\mathcal{B}}(\mathcal{L} \circ \Psi_\mathcal{B} \circ \hat{\phi}_\mathcal{B}) \tag{12}$$

$$+ 3\kappa B \cdot \sqrt{\frac{1}{n_\mathcal{A}^2}\ln\frac{4N_{[]}(\mathcal{P}_{\mathcal{X}_\mathcal{A} \times \mathcal{S}}(\Phi_\mathcal{A}), \frac{1}{n_\mathcal{A}^2})}{\delta}} + \kappa L\sqrt{\frac{2\ln(4/\delta)}{m^2}} + 2L\sqrt{\frac{2\ln(4/\delta)}{n_\mathcal{B}}} \tag{13}$$

Detailed proof of our the Theorem 3.3 in Appendix A.3. Basically, there are three parts on the right side of the oracle inequality. The first part equation 11 is the total variation between the distributions of true $\phi_\mathcal{B}^*$ and $\phi_\mathcal{A}^*$ which represents the common information between modalities $\mathcal{B}, \mathcal{A}$. The error terms $\epsilon_{\mathcal{A}\mathcal{B}}, \epsilon_\mathcal{B}$ are the hyperparameters to control the empirical losses. The second termequation 12 measures the model complexity of the latent variable model $\Phi_\mathcal{B}$ and the prediction model $\Psi_\mathcal{B}$ respectively. Recall that direct ERM of the downstream task in the modality $\mathcal{B}$ is bounded by the complexity of the composition of the latent variable model and the prediction model $\Psi_\mathcal{B} \circ \Phi_\mathcal{B}$ which is much larger than the sum. When $m$ and $n_\mathcal{A}$ are much larger than $n_\mathcal{B}$ which is supposed in our task, the third term equation 13 demonstrates the convergence rate is close to the term in original ERM. Combine all the terms together, we can find that if the total variation between the distributions of true $\phi_\mathcal{B}^*$ and $\phi_\mathcal{A}^*$ is small, the final generalization bound is not worse or even better than the bound of the supervised learning. It indicates that if source and target modalities have more common information or patterns, the algorithm will have a higher probability of distilling more information from the source modality to the downstream task in the target modality. We also validate this observation in experimental results.

## 4 EXPERIMENTS AND DISCUSSION

**Experiments Setup**. To demonstrate the effectiveness of our algorithm, we conduct extensive experiments on various cross-modality tasks.

*Image-sketch*: Here the image is the source modality and the sketch serves as the target modality. Specifically, we use the ImageNet (Deng et al., 2009) as the source dataset $S_\mathcal{A}$ for contrastive learning and Sketchy (Sangkloy et al., 2016) as the paired data $S_{\mathcal{A}\mathcal{B}}$. The downstream recognition task is performed on both Sketchy and TUBerlin datasets to evaluate the generalization performance of the algorithms.

*Video-audio*: Here the video and audio clips become the source and target modalities respectively. We still use ImageNet for contrastive learning on each frame of the video and then distill on a 4.6k subset of VGGSound (Chen et al., 2020a). Random sampled 10k of the rest of the sound data works as the fine-tuning dataset of event recognition in the audio modality.

*Image-depth map*: Here the image and depth map are source and target modalities. ImageNet is used as the source dataset for contrastive learning. We then distill on the NYU-Depth V2 (Silberman et al., 2012). Segmentation is conducted on the depth map only as the downstream task.

| Tasks | image-sketch | | video-audio | image-depth map |
|---|---|---|---|---|
| | Sketchy | TUBerlin | VGGSound | NYU-Dpeth V2 |
| SSL + LE | 64.31 | 54.64 | 18.48 | 13.88 |
| PreSSL + LE | 67.90 | 60.56 | 19.31 | 17.37 |
| CMST + LE | 68.20 | 62.54 | 21.12 | 17.33 |
| CMKD + LE | 70.97 | 64.46 | 26.38 | 17.61 |
| SOCKET + LE | 71.33 | 64.22 | 25.87 | 17.13 |
| CMD + LE (Ours) | 72.61 | 65.70 | **28.30** | 17.36 |
| CMC + LE (Ours) | **73.24** | **68.72** | 28.27 | **18.35** |
| Sup FT | 83.90 | 74.48 | 32.67 | 23.73 |
| SSL + FT | 83.01 | 74.80 | 32.10 | 18.22 |
| PreSSL + FT | 83.75 | 75.50 | 32.37 | 26.06 |
| CMST + FT | 83.32 | 75.36 | 32.11 | 26.16 |
| CMKD + FT | 84.87 | 75.84 | 34.42 | 26.60 |
| SOCKET + FT | 84.93 | 75.48 | 34.15 | 26.33 |
| CMD + FT (Ours) | 85.63 | **77.86** | 35.13 | 27.20 |
| CMC + FT (Ours) | **87.54** | 77.44 | **35.37** | **27.93** |

Table 1: Main results of our method and other baselines on different cross-modality tasks. All the tasks use ResNet50 as the teacher network and ResNet50 as the student network to solve the downstream tasks. We use top-1 accuracy(%) for recognition tasks and mean IoU (%) for the segmentation task. **LE** means a linear evaluation on only the final classification layer; **FT** means fine-tuning the whole network.

We mainly use ResNet (He et al., 2016) models to learn feature extractors. For instance, we may use ResNet50 for ImageNet self-supervised pre-training and distill it to ResNet18/Resnet50 of the target modality. The data augmentation method in SimCLR is used in our contrastive learning and cross-modality distillation. For all training, we use Adam (Kingma & Ba, 2014) optimizer with different learning rates. Commonly, a multistep exponential decay scheduler will be used to adjust the learning rate during training. Detailed settings on the datasets and models can be found in Appendix C.

**Baselines**. In our experiments, we compare our algorithm with the following baselines:

(i). *Sup FT*: Since the downstream task is supervised, we can take the direct training as the baseline. For instance, for the recognition task, we will use the same backbone as our algorithm, and directly minimize the cross-entropy loss. (ii). *SSL*: We also deploy self-supervised learning methods such as SimCLR (Chen et al., 2020b) straightly in the target modality. After the SSL, we may use linear evaluation (LM) or fine-tuning (FT) on the supervised data to get the final performance. (iii). *PreSSL*: In addition to the SSL method, we also test start from the SSL pre-trained model in source modality. In the Image-Sketch task, we can use the SSL pre-trained model of ImageNet as the initial model for SSL of sketch data. (iv). *CMST*: Cross Modality Supervision Transfer (Gupta et al., 2016) is a method that restricts the distance between the mid-level semantic representation of source and target modalities. Specifically, we use the L2 norm to compute the distance between feature extractors of source and target modalities. (v). *CMKD*: In Zhao et al. (2020), they propose to not only add L2 distance on activation maps but also attention maps between the modalities. We adopt the same settings in their work to implement and evaluate this method. (vi). *SOCKET (Ahmed et al., 2022)*: This method does not directly add L2 norm loss on the activation maps but the statistics of the feature maps, e.g., mean and variance.

## 4.1 MAIN RESULTS

**Effectiveness of our algorithm**. As depicted in Table 1, our algorithm surpasses other baselines across various tasks. Specifically, when fine-tuning a one-layer MLP on image-sketch modalities, our method utilizing CMD/CMC loss achieves top-1 accuracies of 72.61%/73.24% on Sketchy, outperforming the best baseline by a margin of 2%. Even when fine-tuning the entire network, our method with CMC loss maintains the best performance, showcasing the effectiveness of our cross-

modality distillation approach. The results obtained on TUBerlin further demonstrate the versatility of the learned features, with our approach achieving the highest accuracies of 68.72%/77.86% through linear evaluation or full fine-tuning.

In addition to image-sketch modalities, our method proves effective in the context of video-audio tasks. Notably, even when utilizing aggregated frame features, our method successfully distills meaningful features from the video/image modality to the audio modality. In the case of linear evaluation with audio data, our method exhibits a 10% advantage over the original SSL method and a 2% margin compared to the best baseline.

Moreover, the results obtained in image-depth segmentation highlight the efficacy of our algorithm. Although our method with linear evaluation or full fine-tuning remains the best approach, the improvement over other baselines is not substantial. This can be attributed to the fact that contrastive learning is primarily designed to discriminate global semantic information among different samples, which might not be as crucial for tasks requiring local semantic information such as segmentation. To address this limitation, future research can explore incorporating contrastive learning on local features. Overall, our algorithm demonstrates superior performance across multiple tasks, including image-sketch, video-audio, and image-depth map segmentation, showcasing its effectiveness and potential for knowledge distillation in cross-modality scenarios.

**Relationship with the generalization bound**. Moreover, our experimental results corroborate the bounds derived in Theorem 3.3. Specifically, we observe that our method exhibits a significant performance improvement through cross-modality distillation in the image-sketch modality, while the improvement in the video-audio modality is more modest. This discrepancy can be attributed to the inherent similarities between images and sketches, which share a common 2-dimensional shape representation. Sketches can be viewed as a direct abstraction of RGB images, leading to a more effective knowledge transfer. On the other hand, video and audio data exhibit larger variations. During training, audio is typically represented by spectrograms, while video is treated as a sequence of images. Consequently, the performance improvement achieved by our method in the video-audio task amounts to less than a 2% improvement. Conversely, in the image-sketch modalities, our method demonstrates a notable boost in performance, ranging from 2% to 3% improvement, even when starting from a higher baseline accuracy. These results just validate the insight in our theoretical results.

## 4.2 MORE DISCUSSION ABOUT THE METHOD

| Models | ResNet50(1x) | ResNet50(2x) | ResNet50(4x) |
|---|---|---|---|
| ResNet18 | 67.06 | 67.64 | 67.34 |
| ResNet50 | 68.72 | 68.90 | 69.58 |

Table 2: The results of image-sketch task with different teacher and student networks. We report the top-1 accuracy of downstream classification on the TUBerlin sketch with linear evaluation.

**Distillation with more structures**. Though the modality difference can be seen as the main distillation source in cross-modality distillation, it is still natural to consider distilling a large model into a small model in this case. Thus, in this part, we mainly study how model sizes or structures affect the performance of our method and illustrate that our method can also be used even when distilling the large teacher to a small student model. In detail, we test our algorithm with ResNet50 (1x/2x/4x) as teacher nets and ResNet18, ResNet50 as the student nets. Here, the ResNet50 (1x/2x/4x) means the original ResNet50 with different widths which is the same setting in Chen et al. (2020b). From Table 2, we can find when the teacher model size becomes larger the distilled student models will have a better performance which just fits with the classical results in knowledge distillation. It shows our algorithm's ability to distill information captured not only by the modality difference but also by the model structure inductive bias.

**Distillation with less numbers of samples**. We further test our algorithm with less paired data when cross-modality distilling. We take $m$ as the number of paired data used in distillation and $M$ as the number of whole paired data. On both Sketchy and TUBerlin, as shown in Figure 2, our method works well even when the paired data reduces to $m/M = 20\%$ of the whole samples.

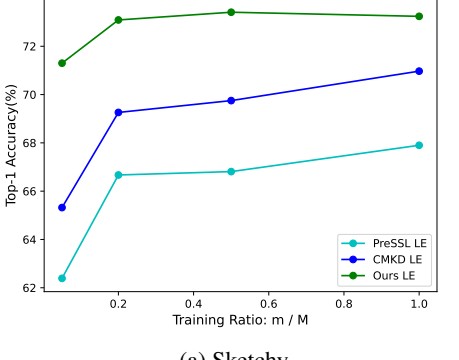

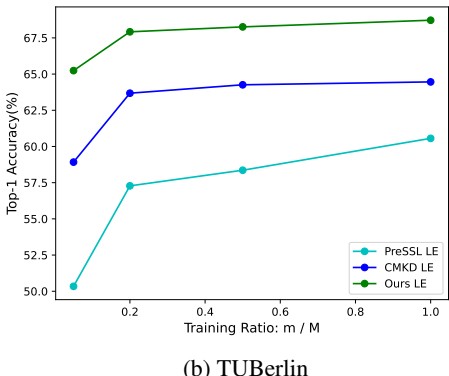

(a) Sketchy                                      (b) TUBerlin

Figure 2: The results of image-sketch tasks with different numbers of distilling samples. The $m/M$ means the percentage of numbers used in distillation. We report the top-1 accuracy of downstream classification on the Sketchy and TUBerlin.

| Models | $n = 1$ | $n = 5$ | $n = 10$ | Standard(n=60) |
|---|---|---|---|---|
| CMKD LE | 10.21 | 33.18 | 46.27 | 64.46 |
| Ours LE | 19.19 | 42.58 | 52.11 | 68.72 |
| Sup FT | 19.36 | 42.92 | 50.43 | 74.48 |
| CMKD FT | 19.22 | 43.19 | 54.11 | 75.84 |
| Ours FT | 20.29 | 46.53 | 59.90 | 77.44 |

Table 3: The results of image-sketch task with different fine-tuning samples, where $n$=1/5/10 means the number of training samples in each class. We report the top-1 accuracy of downstream classification on the TUBerlin sketch.

Notably, when we decrease the entire training data to only 5% of the original training setting, our method experiences a marginal drop of approximately 3%. In contrast, the performances of CMKD and PreSSL methods decline as the number of training samples decreases, and they exhibit a rapid drop when only 5% of the data remains. These results highlight the resilience of our algorithm when working with a limited number of paired data for cross-modality distillation.

**Downstream tasks under few-shot setting**. To better show that our method can learn meaningful semantic information by cross-modality distillation, we use a few-shot setting test that is used in many self-supervised learning works He et al. (2020). Specifically, we test our algorithm on image-sketch modalities and fine-tuning on TUBerlin with each class consisting of only 1,5,10 samples. It can be observed in Table 3 that when $n$ is small our method even with linear evaluation can achieve comparable performance to the supervised full fine-tuning and our methods just outperforms a lot when we use full fine-tuning as well. This indicates that our method can distill efficient semantic features from the source modality.

## 5 CONCLUSION

In this paper, we design a cross-modality contrastive distillation (CMCD) framework for transferring generalizable features from a major source modality to a minor target modality. The proposed framework leverages contrastive learning to fully investigate the positive and negative relationships behind the paired data. Comprehensive experiments covering various modalities (e.g., images, sketches, depth maps, videos, and audio) and tasks (e.g., recognition and segmentation) shows that our algorithms outperform other methods consistently. Furthermore, we provide a convergence analysis that reveals the test error in the target modality will be bounded by the distance between source and target modalities in our algorithm. These findings underscore the effectiveness and versatility of CMCD as a means of achieving robust feature transfer in various real-world scenarios.

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

# A   DETAILED PROOF OF THE LEMMAS AND THEOREMS

## A.1   PROOF OF THE THEOREM 3.1

*Proof.* We just following the proof in the Ge et al. (2023). First, we reformulate the optimization problem in equation 4 as

$$\hat{\phi}_{\mathcal{A}} = \arg\max_{\phi \in \Phi_{\mathcal{A}}} \sum_{i,j=1}^{n_{\mathcal{A}}} \log p_\phi(\phi(\boldsymbol{x}_{ij}^{\mathcal{A}}), s_{ij}) \tag{14}$$

where $\phi(\boldsymbol{x}_{ij}) = (z_i^{\mathcal{A}}, z_j^{\mathcal{A}}) = (\phi(x_i^{\mathcal{A}}), \phi(x_j^{\mathcal{A}}))$, and

$$p_\phi(\phi(\boldsymbol{x}_{ij}^{\mathcal{A}}), s_{ij}) = \frac{\exp(z_i^{\mathcal{A}} \cdot z_j^{\mathcal{A}}/\tau)}{\sum_t \exp(z_t^{\mathcal{A}} \cdot z_j^{\mathcal{A}}/\tau)} \tag{15}$$

the Gibbs distribution for the paired data. In fact, it just formulates the cross-modality contrastive learning framework by the maximum likelihood estimation (MLE). We ignore the $\mathcal{A}$ subscripts/upscripts and the side information $s$ for notation simplicity in the proof. By the definition of $\hat{\phi}$, we have

$$0 \le \frac{1}{2}\left(\sum_{i,j=1}^n \log p_{\hat{\phi}}(\boldsymbol{x}_{ij}) - \sum_{i,j=1}^n \log p_{\phi^*}(\boldsymbol{x}_{ij})\right) \tag{16}$$

$$= \frac{1}{2}\sum_{i,j=1}^n \log \frac{p_{\hat{\phi}}(\boldsymbol{x}_{ij})}{p_{\phi^*}(\boldsymbol{x}_{ij})} \tag{17}$$

$$\tag{18}$$

To construct the relationship between $d_{TV}$ and the previous formula, we use Markov inequality and Boole inequality (subadditivity of events). Recall that we define $\mathcal{P}_{\mathcal{X} \times \mathcal{S}}(\Phi) = \{p_\phi(\boldsymbol{x}, s)|\phi \in \Phi\}$ as the possible distribution family of $\Phi$. For notation simplicity, we denote $\mathcal{P}_{\mathcal{X}_{\mathcal{A}} \times \mathcal{S}}(\Phi_{\mathcal{A}})$ as $\mathcal{P}$ in this proof. Then we denote the $\epsilon$-bracket class as $\mathcal{N}_{[]}(\mathcal{P}, \epsilon)$, $N_{[]}(\mathcal{P}, \epsilon) = |\mathcal{N}_{[]}(\mathcal{P}, \epsilon)|$. For any $\overline{p}_\phi \in \mathcal{N}_{[]}(\mathcal{P}, \epsilon)$, we have the following Markov inequality,

$$\mathbb{P}(\exp(\frac{1}{2}\sum_{i,j=1}^n \log \frac{\overline{p}_\phi(\boldsymbol{x}_{ij})}{p_{\phi^*}(\boldsymbol{x}_{ij})} \ge t)) \le \frac{\mathbb{E}[\exp(\frac{1}{2}\sum_{i,j=1}^n \log \frac{\overline{p}_{\hat{\phi}}(\boldsymbol{x}_{ij})}{p_{\phi^*}(\boldsymbol{x}_{ij})})]}{t} \tag{19}$$

$$\mathbb{P}\left(\exp\left(\frac{1}{2}\sum_{i,j=1}^n \log \frac{\overline{p}_\phi(\boldsymbol{x}_{ij})}{p_{\phi^*}(\boldsymbol{x}_{ij})} \ge \frac{C\mathbb{E}[\exp(\frac{1}{2}\sum_{i,j=1}^n \log \frac{\overline{p}_{\hat{\phi}}(\boldsymbol{x}_{ij})}{p_{\phi^*}(\boldsymbol{x}_{ij})})]}{\delta}\right)\right) \le \delta/C \tag{20}$$

$$\tag{21}$$

Define the event $D_{\overline{p}_\phi}$ as

$$D_{\overline{p}_\phi} = \{\boldsymbol{x} : \exp\left(\frac{1}{2}\sum_{i,j=1}^n \log \frac{\overline{p}_\phi(\boldsymbol{x}_{ij})}{p_{\phi^*}(\boldsymbol{x}_{ij})}\right) \ge \frac{C\mathbb{E}[\exp(\frac{1}{2}\sum_{i,j=1}^n \log \frac{\overline{p}_\phi(\boldsymbol{x}_{ij})}{p_{\phi^*}(\boldsymbol{x}_{ij})})]}{\delta}\} \tag{22}$$

Then by iterating over all $\overline{p}_\phi \in \mathcal{N}_{[]}(\mathcal{P}_A, \epsilon)$ we have,

$$\mathbb{P}(\cup_{\overline{p}_\phi \in \mathcal{N}_{[]}(\mathcal{P}_A,\epsilon)} D_{\overline{p}_\phi}) \le \sum_{\overline{p}_\phi \in \mathcal{N}_{[]}(\mathcal{P}_A,\epsilon)} \mathbb{P}(D_{\overline{p}_\phi}) \tag{23}$$

$$\le \frac{N_{[]}(\mathcal{P}_A, \epsilon) \cdot \delta}{C} \tag{24}$$

Take $C = N_{[]}(\mathcal{P}, \epsilon)$, we have with probability at least $1 - \delta$, for all $\overline{p}_\phi \in \mathcal{N}_{[]}(\mathcal{P}, \epsilon)$

$$\exp\left(\frac{1}{2}\sum_{i,j=1}^n \log \frac{\overline{p}_\phi(\boldsymbol{x}_{ij})}{p_{\phi^*}(\boldsymbol{x}_{ij})}\right) \le \mathbb{E}[\exp(\frac{1}{2}\sum_{i,j=1}^n \log \frac{\overline{p}_\phi(\boldsymbol{x}_{ij})}{p_{\phi^*}(\boldsymbol{x}_{ij})})] \cdot \frac{N_{[]}(\mathcal{P}, \epsilon)}{\delta} \tag{25}$$

$$\frac{1}{2}\sum_{i,j=1}^n \log \frac{\overline{p}_\phi(\boldsymbol{x}_{ij})}{p_{\phi^*}(\boldsymbol{x}_{ij})} \le \log \mathbb{E}[\exp(\frac{1}{2}\sum_{i,j=1}^n \log \frac{\overline{p}_\phi(\boldsymbol{x}_{ij})}{p_{\phi^*}(\boldsymbol{x}_{ij})})] + \log \frac{N_{[]}(\mathcal{P}, \epsilon)}{\delta} \tag{26}$$

By the definition of bracket class, $\overline{p}_{\hat{\phi}}$ satisfies, with probability at least $1 - \delta$

$$0 \le \frac{1}{2} \sum_{i,j=1}^{n} \log \frac{\overline{p}_{\hat{\phi}}(\boldsymbol{x}_{ij})}{p_{\phi^*}(\boldsymbol{x}_{ij})} \le \log \mathbb{E}[\exp(\frac{1}{2} \sum_{i,j=1}^{n} \log \frac{\overline{p}_{\hat{\phi}}(\boldsymbol{x}_{ij})}{p_{\phi^*}(\boldsymbol{x}_{ij})})] + \log \frac{N_{[]}(\mathcal{P}, \epsilon)}{\delta} \tag{27}$$

$$= \sum_{i,j=1}^{n} \log \mathbb{E}[\sqrt{\frac{\overline{p}_{\hat{\phi}}(\boldsymbol{x}_{ij})}{p_{\phi^*}(\boldsymbol{x}_{ij})}}] + \log \frac{N_{[]}(\mathcal{P}, \epsilon)}{\delta} \tag{28}$$

$$= m^2 \log \int \sqrt{\overline{p}_{\hat{\phi}}(\boldsymbol{x}) \cdot p_{\phi^*}(\boldsymbol{x})} d\boldsymbol{x} + \log \frac{N_{[]}(\mathcal{P}, \epsilon)}{\delta} \tag{29}$$

$$\le m^2 (\int \sqrt{\overline{p}_{\hat{\phi}}(\boldsymbol{x}) \cdot p_{\phi^*}(\boldsymbol{x})} d\boldsymbol{x} - 1) + \log \frac{N_{[]}(\mathcal{P}, \epsilon)}{\delta} \tag{30}$$

$$\tag{31}$$

By rearranging the terms,

$$1 - \int \sqrt{\overline{p}_{\hat{\phi}}(\boldsymbol{x}) \cdot p_{\phi^*}(\boldsymbol{x})} d\boldsymbol{x} \le \frac{1}{m^2} \log \frac{N_{[]}(\mathcal{P}_A, \epsilon)}{\delta} \tag{32}$$

$$\int \left( \sqrt{\overline{p}_{\hat{\phi}}(\boldsymbol{x})} - \sqrt{p_{\phi^*}(\boldsymbol{x})} \right)^2 d\boldsymbol{x} \le \frac{2}{m^2} \log \frac{N_{[]}(\mathcal{P}_A, \epsilon)}{\delta} \tag{33}$$

$$\tag{34}$$

By the definition of $\epsilon$-bracket class, we have

$$\int \left( \sqrt{\overline{p}_{\hat{\phi}}(\boldsymbol{x})} + \sqrt{p_{\phi^*}(\boldsymbol{x})} \right)^2 dx \le 2 + 2 \int \sqrt{\overline{p}_{\hat{\phi}}(\boldsymbol{x}) \cdot p_{\phi^*}(\boldsymbol{x})} dx \tag{35}$$

$$\le 2 + \int \overline{p}_{\hat{\phi}}(\boldsymbol{x}) + p_{\phi^*}(\boldsymbol{x}) dx \tag{36}$$

$$\le 2 + 2(\epsilon + 1) = 2\epsilon + 4 \tag{37}$$

Now we can bound the $d_{TV}$ by Cauchy-Schwarz inequality, with probability at least $1 - \delta$

$$d_{TV} \left( \mathbb{P}_{\hat{\phi}}(\boldsymbol{x}), \mathbb{P}_{\phi^*}(\boldsymbol{x}) \right) = \frac{1}{2} \int |p_{\hat{\phi}}(\boldsymbol{x}) - p_{\phi^*}(\boldsymbol{x})| d\boldsymbol{x} \tag{38}$$

$$\le \frac{1}{2} \int |\overline{p}_{\hat{\phi}}(\boldsymbol{x}) - p_{\phi^*}(\boldsymbol{x})| d\boldsymbol{x} + \frac{1}{2} \int |p_{\hat{\phi}}(\boldsymbol{x}) - \overline{p}_{\hat{\phi}}(\boldsymbol{x})| d\boldsymbol{x} \tag{39}$$

$$\le \frac{1}{2} \left( \int \left( \sqrt{\overline{p}_{\hat{\phi}}(\boldsymbol{x})} - \sqrt{p_{\phi^*}(\boldsymbol{x})} \right)^2 d\boldsymbol{x} \cdot \int \left( \sqrt{\overline{p}_{\hat{\phi}}(\boldsymbol{x})} + \sqrt{p_{\phi^*}(\boldsymbol{x})} \right)^2 d\boldsymbol{x} \right)^{1/2} + \frac{\epsilon}{2} \tag{40}$$

$$\le \frac{1}{2} \sqrt{\frac{2}{n^2} \log \frac{N_{[]}(\mathcal{P}, \epsilon)}{\delta} \cdot (2\epsilon + 4)} + \frac{\epsilon}{2} \tag{41}$$

set $\epsilon = \frac{1}{m^2}$ we can bound the formula above by

$$d_{TV} \left( \mathbb{P}_{\hat{\phi}}(\boldsymbol{x}), \mathbb{P}_{\phi^*}(\boldsymbol{x}) \right) \le 3 \sqrt{\frac{1}{n^2} \log \frac{N_{[]}(\mathcal{P}, \frac{1}{n^2})}{\delta}} \tag{42}$$

$$\square$$

## A.2 Proof of the Theorem 3.2

*Proof.* The proof is mainly from Chap. 6 in Zhang (2023).
First, we define

$$\epsilon(\mathcal{L} \circ \Phi_{\mathcal{B}}, S_m^2) = \sup_{\phi_{\mathcal{B}} \in \Phi_{\mathcal{B}}} [\mathbb{E}[\mathcal{L}(\hat{\phi}_{\mathcal{A}}, \phi_{\mathcal{B}}, \boldsymbol{x}, s)] - \frac{1}{m^2} \sum_{i,j=1}^{m} \mathcal{L}(\hat{\phi}_{\mathcal{A}}, \phi_{\mathcal{B}}, \boldsymbol{x}_{ij}, s_{ij})] \tag{43}$$

and

$$\epsilon_n(\mathcal{L} \circ \Phi_{\mathcal{B}}) = \mathbb{E}_{S_m^2} \epsilon(\mathcal{L} \circ \Phi_{\mathcal{B}}, S_m^2) \tag{44}$$

where $S_m^2 = \{(x_i^{\mathcal{A}}, x_i^{\mathcal{B}})\}_m \times \{(x_i^{\mathcal{A}}, x_i^{\mathcal{B}})\}_m$. Then by the symmetrization, we can prove

$$\epsilon_n(\mathcal{L} \circ \Phi_{\mathcal{B}}) \leq 2R_n(\mathcal{L} \circ \Phi_{\mathcal{B}}) \tag{45}$$

we do not give detailed proof, readers can refer to Theorem 6.3 in Zhang (2023). Consider $f(\boldsymbol{x}_{11}, \ldots, \boldsymbol{x}_{mm}) = \epsilon(\mathcal{L} \circ \Phi_{\mathcal{B}}, S_m^2)$, by the assumption that $\sup_{\phi_{\mathcal{B}} \in \Phi_{\mathcal{B}}, \boldsymbol{x}_{ij}} \langle \phi_{\mathcal{B}}(\boldsymbol{x}_i), \phi_{\mathcal{B}}(\boldsymbol{x}_j) \rangle \leq B$, we can check the condition for McDiarmid's inequality,

$$\sup_{\boldsymbol{x}_{11}, \ldots, \boldsymbol{x}_{mm}, \boldsymbol{x}_{ij}'} |f(\boldsymbol{x}_{11}, \ldots, \boldsymbol{x}_{ij}', \ldots, \boldsymbol{x}_{mm}) - f(\boldsymbol{x}_{11}, \ldots, \boldsymbol{x}_{ij}', \ldots, \boldsymbol{x}_{mm})| \tag{46}$$

$$\leq \frac{1}{m^2} \sup_{\boldsymbol{x}_{ij}, \boldsymbol{x}_{ij}'} |\sup_{\phi_{\mathcal{B}}} [\mathcal{L}(\hat{\phi}_{\mathcal{A}}, \phi_{\mathcal{B}}, \boldsymbol{x}_{ij}', s_{ij}') - \mathcal{L}(\hat{\phi}_{\mathcal{A}}, \phi_{\mathcal{B}}, \boldsymbol{x}_{ij}, s_{ij})]| \tag{47}$$

$$\leq \frac{1}{m^2} \sup_{\boldsymbol{x}_{ij}, \boldsymbol{x}_{ij}'} \sup_{\phi_{\mathcal{B}}} |\mathcal{L}(\hat{\phi}_{\mathcal{A}}, \phi_{\mathcal{B}}, \boldsymbol{x}_{ij}', s_{ij}') - \mathcal{L}(\hat{\phi}_{\mathcal{A}}, \phi_{\mathcal{B}}, \boldsymbol{x}_{ij}, s_{ij})| \tag{48}$$

$$\leq \frac{1}{m^2} 2 |\log \frac{1 + \exp(B)}{1 + \exp(-B)}| \tag{49}$$

$$= \frac{2B}{m^2} \tag{50}$$

$$\tag{51}$$

then we can apply McDiarmid's inequality,

$$\mathbb{P}(f(\boldsymbol{x}_{11}, \ldots, \boldsymbol{x}_{mm}) \geq \mathbb{E}_{S_m^2} f(\boldsymbol{x}_{11}, \ldots, \boldsymbol{x}_{mm}) + \epsilon) \leq \exp(\frac{-m^2 \epsilon^2}{2B^2}) \tag{52}$$

then with probability at least $1 - \delta$,

$$f(\boldsymbol{x}_{11}, \ldots, \boldsymbol{x}_{mm}) \leq \mathbb{E}_{S_m^2} f(\boldsymbol{x}_{11}, \ldots, \boldsymbol{x}_{mm}) + B \sqrt{\frac{2 \ln(1/\delta)}{m^2}} \tag{53}$$

$$\sup_{\phi_{\mathcal{B}} \in \Phi_{\mathcal{B}}} [\mathbb{E}[\mathcal{L}(\hat{\phi}_{\mathcal{A}}, \phi_{\mathcal{B}}, \boldsymbol{x}, s)] - \frac{1}{m^2} \sum_{i,j=1}^{m} \mathcal{L}(\hat{\phi}_{\mathcal{A}}, \phi_{\mathcal{B}}, \boldsymbol{x}_{ij}, s_{ij})] \leq \epsilon_n(\mathcal{L} \circ \Phi_{\mathcal{B}}) + B \sqrt{\frac{2 \ln(1/\delta)}{m^2}} \tag{54}$$

Combined with the result of equation 45, we have with probability at least $1 - \delta$, for any $\phi_{\mathcal{B}} \in \Phi_{\mathcal{B}}$,

$$\mathbb{E}[\mathcal{L}(\hat{\phi}_{\mathcal{A}}, \hat{\phi}_{\mathcal{B}}, \boldsymbol{x}, s)] - \frac{1}{m^2} \sum_{i,j=1}^{m} \mathcal{L}(\hat{\phi}_{\mathcal{A}}, \hat{\phi}_{\mathcal{B}}, \boldsymbol{x}_{ij}, s_{ij}) \leq 2R_n(\mathcal{L} \circ \Phi_{\mathcal{B}}) + B \sqrt{\frac{2 \ln(1/\delta)}{m^2}} \tag{55}$$

A similar discussion shows that with probability at least $1 - \delta$, $\phi_{\mathcal{B}} \in \Phi_{\mathcal{B}}$,

$$\frac{1}{m^2} \sum_{i,j=1}^{m} \mathcal{L}(\hat{\phi}_{\mathcal{A}}, \hat{\phi}_{\mathcal{B}}, \boldsymbol{x}_{ij}, s_{ij}) - \mathbb{E}[\mathcal{L}(\hat{\phi}_{\mathcal{A}}, \hat{\phi}_{\mathcal{B}}, \boldsymbol{x}, s)] \leq 2R_n(\mathcal{L} \circ \Phi_{\mathcal{B}}) + B \sqrt{\frac{2 \ln(1/\delta)}{m^2}} \tag{56}$$

$$\square$$

## A.3 PROOF OF THE THEOREM 3.3

We first introduce a lemma.

**Lemma A.1** (Bound of ERM.). *Suppose that $\mathcal{L}(\cdot, \cdot)$ is a $L$-bounded loss function. Given a fixed $\phi \in \Phi$, with probability at least $1 - \delta$, for any $\psi \in \Psi$,*

$$\mathbb{E}[\mathcal{L}(\psi \circ \phi(x), y)] - \frac{1}{n} \sum_{i=1}^{n} \mathcal{L}(\psi \circ \phi(x), y) \leq 2R_n(\mathcal{L} \circ \Psi \circ \phi) + L \sqrt{\frac{2 \ln(1/\delta)}{n}} \tag{57}$$

*Proof.* This proof is almost the same as the proof in Theorem 3.2. First, we define $\epsilon(\mathcal{L} \circ \Psi \circ \phi, S_n) = \sup_{\psi \in \Psi} [\mathbb{E}[\mathcal{L}(\psi \circ \phi(x), y)] - \frac{1}{n} \sum_{i=1}^{n} \mathcal{L}(\psi \circ \phi(x), y)]$ and $\epsilon_n(\mathcal{L} \circ \Psi \circ \phi) = \mathbb{E}_{S_n} \epsilon(\mathcal{L} \circ \Psi \circ \phi, S_n)$ where $S_n = \{(x_i, y_i)\}_n$. Then by the symmetrization, we can prove

$$\epsilon_n(\mathcal{L} \circ \Psi \circ \phi) \leq 2R_n(\mathcal{L} \circ \Psi \circ \phi) \tag{58}$$

we do not give detailed proof, readers can refer to Theorem 6.3 in Zhang (2023). Consider $f(X_1, \ldots, X_n) = \sup_{\psi \in \Psi} [\mathbb{E}[\mathcal{L}(\psi \circ \phi(x), y)] - \frac{1}{n} \sum_{i=1}^{n} \mathcal{L}(\psi \circ \phi(x), y)]$, it is obvious that $\sup_{x_1, \ldots, x_n, x_i'} |f(x_1, \ldots, x_i, \ldots, x_n) - f(x_1, \ldots, x_i', \ldots, x_n)| \leq \frac{2}{n} L$, then we can apply McDiarmid's inequality,

$$\mathbb{P}(f(x_1, \ldots, x_n) \geq \mathbb{E}_{S_n} f(x_1, \ldots, x_n) + \epsilon) \leq \exp(\frac{-n\epsilon^2}{2L^2}) \tag{59}$$

then with probability at least $1 - \delta$,

$$f(x_1, \ldots, x_n) \leq \mathbb{E}_{S_n} f(x_1, \ldots, x_n) + L\sqrt{\frac{2\ln(1/\delta)}{n}} \tag{60}$$

$$\sup_{\psi \in \Psi} [\mathbb{E}[\mathcal{L}(\psi \circ \phi(x), y)] - \frac{1}{n} \sum_{i=1}^{n} \mathcal{L}(\psi \circ \phi(x), y)] \leq \epsilon_n(\mathcal{L} \circ \Psi \circ \phi) + L\sqrt{\frac{2\ln(1/\delta)}{n}} \tag{61}$$

Combined with the result of equation 45, we have with probability at least $1 - \delta$, for any $\psi \in \Psi$,

$$\mathbb{E}[\mathcal{L}(\psi \circ \phi(x), y)] - \frac{1}{n} \sum_{i=1}^{n} \mathcal{L}(\psi \circ \phi(x), y) \leq 2R_n(\mathcal{L} \circ \Psi \circ \phi) + L\sqrt{\frac{2\ln(1/\delta)}{n}} \tag{62}$$

A similar discussion shows that with probability at least $1 - \delta$, for any $\psi \in \Psi$,

$$\frac{1}{n} \sum_{i=1}^{n} \mathcal{L}(\psi \circ \phi(x), y) - \mathbb{E}[\mathcal{L}(\psi \circ \phi(x), y)] \leq 2R_n(\mathcal{L} \circ \Psi \circ \phi) + L\sqrt{\frac{2\ln(1/\delta)}{n}} \tag{63}$$

take $1 - \delta/2$ for each inequality and combine the results, we get with probability at least $1 - \delta$, for any $\psi \in \Psi$,

$$|\mathbb{E}[\mathcal{L}(\psi \circ \phi(x), y)] - \frac{1}{n} \sum_{i=1}^{n} \mathcal{L}(\psi \circ \phi(x), y)| \leq 2R_n(\mathcal{L} \circ \Psi \circ \phi) + L\sqrt{\frac{2\ln(2/\delta)}{n}} \tag{64}$$

□

Now we prove the Theorem 3.3,

*Proof.* The proof starts from the standard convergence analysis with Rademacher complexity. By the Lemma A.1, given the fixed $\hat{\phi}_\mathcal{B}(x)$, we have with probability at least $1 - \delta$,

$$\mathbb{E}[\mathcal{L}(\hat{\psi}_\mathcal{B} \circ \hat{\phi}_\mathcal{B}(x), y)] \leq \frac{1}{n} \sum_{i=1}^{n_\mathcal{B}} \mathcal{L}(\hat{\psi}_\mathcal{B} \circ \hat{\phi}_\mathcal{B}(x), y) + 2R_n(\mathcal{L} \circ \Psi_\mathcal{B} \circ \hat{\phi}_\mathcal{B}) + L\sqrt{\frac{2\ln(1/\delta)}{n_\mathcal{B}}} \tag{65}$$

we only need to handle the empirical risk $\frac{1}{n} \sum_{i=1}^{n_\mathcal{B}} \mathcal{L}(\hat{\phi}_\mathcal{B}, \hat{\psi}_\mathcal{B}(x), y)$, from the definition of $\hat{\psi}_\mathcal{B}$ in equation 6 we get,

$$\frac{1}{n} \sum_{i=1}^{n_\mathcal{B}} \mathcal{L}(\hat{\psi}_\mathcal{B} \circ \hat{\phi}_\mathcal{B}(x), y) \leq \epsilon_\mathcal{B} + \frac{1}{n} \sum_{i=1}^{n_\mathcal{B}} \mathcal{L}(\psi_\mathcal{B}^* \circ \hat{\phi}_\mathcal{B}(x), y) - \mathbb{E}[\mathcal{L}(\psi_\mathcal{B}^* \circ \hat{\phi}_\mathcal{B}(x), y)] \tag{66}$$

$$+ \mathbb{E}[\mathcal{L}(\psi_\mathcal{B}^* \circ \hat{\phi}_\mathcal{B}(x), y)] \tag{67}$$

the first term equation 66 can be bounded by concentration inequality and we only need to bound the second term equation 67 further. By the assumption 3.1,

$$\mathbb{E}[\mathcal{L}(\hat{\phi}_{\mathcal{B}}, \psi_{\mathcal{B}}^*(x), y)] \le \kappa \mathbb{E}[\mathbb{E}_{x'}[\mathcal{L}_{\text{CMD}}(\hat{\phi}_{\mathcal{B}}, \phi_{\mathcal{B}}^*, (x, x'), s)]] \tag{68}$$

$$= \kappa \mathbb{E}[\mathcal{L}_{\text{CMD}}(\hat{\phi}_{\mathcal{B}}, \phi_{\mathcal{B}}^*, (x, x'), s)] \quad (\text{denote } \boldsymbol{x} = (x, x')) \tag{69}$$

$$= \kappa(\mathbb{E}[\mathcal{L}_{\text{CMD}}(\hat{\phi}_{\mathcal{B}}, \phi_{\mathcal{B}}^*, \boldsymbol{x}, s)] - \mathbb{E}[\mathcal{L}_{\text{CMD}}(\hat{\phi}_{\mathcal{B}}, \phi_{\mathcal{A}}^*, \boldsymbol{x}, s)] \tag{70}$$

$$+ \mathbb{E}[\mathcal{L}_{\text{CMD}}(\hat{\phi}_{\mathcal{B}}, \phi_{\mathcal{A}}^*, \boldsymbol{x}, s)] - \mathbb{E}[\mathcal{L}_{\text{CMD}}(\hat{\phi}_{\mathcal{B}}, \hat{\phi}_{\mathcal{A}}, \boldsymbol{x}, s)] \tag{71}$$

$$+ \mathbb{E}[\mathcal{L}_{\text{CMD}}(\hat{\phi}_{\mathcal{B}}, \hat{\phi}_{\mathcal{A}}, \boldsymbol{x}, s)]) \tag{72}$$

$$= \kappa(\mathbb{E}[-p_{\phi_{\mathcal{B}}^*}(\boldsymbol{x}, s) \log p_{\hat{\phi}_{\mathcal{B}}}(\boldsymbol{x}, s) + p_{\phi_{\mathcal{A}}^*}(\boldsymbol{x}, s) \log p_{\hat{\phi}_{\mathcal{B}}}(\boldsymbol{x}, s)] \tag{73}$$

$$+ \mathbb{E}[-p_{\phi_{\mathcal{A}}^*}(\boldsymbol{x}, s) \log p_{\hat{\phi}_{\mathcal{B}}}(\boldsymbol{x}, s) + p_{\hat{\phi}_{\mathcal{A}}}(\boldsymbol{x}, s) \log p_{\hat{\phi}_{\mathcal{B}}}(\boldsymbol{x}, s)] \tag{74}$$

$$+ \mathbb{E}[\mathcal{L}_{\text{CMD}}(\hat{\phi}_{\mathcal{B}}, \hat{\phi}_{\mathcal{A}}, \boldsymbol{x}, s)]) \tag{75}$$

$$\le \kappa B \cdot \left( d_{TV}(\mathbb{P}_{\phi_{\mathcal{B}}^*}, \mathbb{P}_{\phi_{\mathcal{A}}^*}) + d_{TV}(\mathbb{P}_{\hat{\phi}_{\mathcal{A}}}, \mathbb{P}_{\phi_{\mathcal{A}}^*}) \right) \tag{76}$$

$$+ \kappa \mathbb{E}[\mathcal{L}_{\text{CMD}}(\hat{\phi}_{\mathcal{B}}, \hat{\phi}_{\mathcal{A}}, \boldsymbol{x}, s)] \tag{77}$$

$$\tag{78}$$

where the inequality equation 104 comes from a same argument as equation 49.

To derive the final result, define two events,

$$D_{\mathcal{A}} = \left\{ S_n^{\mathcal{A}} : d_{TV}(\mathbb{P}_{\hat{\phi}_{\mathcal{A}}}(\boldsymbol{x}, s), \mathbb{P}_{\phi_{\mathcal{A}}^*}(\boldsymbol{x}, s)) \le 3\sqrt{\frac{1}{n_{\mathcal{A}}^2} \log \frac{N_{[]}(\mathcal{P}_{\mathcal{X}_{\mathcal{A}} \times \mathcal{S}}(\Phi_{\mathcal{A}}), \frac{1}{n_{\mathcal{A}}^2})}{\delta}} \right\} \tag{79}$$

$$D_{\mathcal{A}\mathcal{B}} = \left\{ S_m^2 : \mathbb{E}[\mathcal{L}(\hat{\phi}_{\mathcal{B}}, \hat{\phi}_{\mathcal{A}}, \boldsymbol{x}, s)] - \frac{1}{m^2} \sum_{i,j=1}^m \mathcal{L}_{\text{CMD}}(\hat{\phi}_{\mathcal{B}}, \hat{\phi}_{\mathcal{A}}, \boldsymbol{x}_{ij}, s_{ij}) \le 2R_n(\mathcal{L}_{\text{CMD}} \circ \Phi_{\mathcal{B}}) + L\sqrt{\frac{2\ln(1/\delta)}{m^2}} \right\} \tag{80}$$

By the Theorem 3.2 and Lemma 3.1, we have $\mathbb{P}(D_{\mathcal{A}}) \ge 1 - \delta, \mathbb{P}(D_{\mathcal{A}\mathcal{B}}|\hat{\phi}_{\mathcal{A}}) \ge 1 - \delta$, then consider the $\mathbb{P}(D_{\mathcal{A}} \cap D_{\mathcal{A}\mathcal{B}})$,

$$\mathbb{P}(D_{\mathcal{A}} \cap D_{\mathcal{A}\mathcal{B}}) = \mathbb{E}[\mathbb{1}_{D_{\mathcal{A}}} \mathbb{P}(D_{\mathcal{A}\mathcal{B}}|\hat{\phi}_{\mathcal{A}})] \tag{81}$$

$$\ge (1 - \delta) \cdot \mathbb{P}(D_{\mathcal{A}}) \tag{82}$$

$$\ge (1 - \delta)^2 \ge 1 - 2\delta \tag{83}$$

So with probability at least $1 - \delta$,

$$\mathbb{E}[\mathcal{L}(\hat{\phi}_{\mathcal{B}}, \psi_{\mathcal{B}}^*(x), y)] \le \kappa B \cdot d_{TV}(p_{\phi_{\mathcal{B}}^*}, p_{\phi_{\mathcal{A}}^*}) \tag{84}$$

$$+ 3\kappa B \cdot \sqrt{\frac{1}{n_{\mathcal{A}}^2} \ln \frac{2N_{[]}(\mathcal{P}_{\mathcal{X}_{\mathcal{A}} \times \mathcal{S}}(\Phi_{\mathcal{A}}), \frac{1}{n_{\mathcal{A}}^2})}{\delta}} \tag{85}$$

$$+ \kappa(\epsilon_{\mathcal{A}\mathcal{B}} + 2R_{m^2}(\mathcal{L}_{\text{CMD}} \circ \Phi_{\mathcal{B}}) + L\sqrt{\frac{2\ln(2/\delta)}{m^2}}) \tag{86}$$

By Chernoff bound, with probability at least $1 - \delta$, we have

$$\frac{1}{n_{\mathcal{B}}} \sum_{i=1}^{n_{\mathcal{B}}} \mathcal{L}(\hat{\phi}_{\mathcal{B}}, \psi_{\mathcal{B}}^*(x), y) - \mathbb{E}[\mathcal{L}(\hat{\phi}_{\mathcal{B}}, \psi_{\mathcal{B}}^*(x), y)] \le L\sqrt{\frac{2\ln(1/\delta)}{n_{\mathcal{B}}}} \tag{87}$$

Take $1 - \delta/2$ for equation 65 and equation 87, we get with probability $1 - \delta$,

$$\mathbb{E}[\mathcal{L}(\hat{\phi}_{\mathcal{B}}, \hat{\psi}_{\mathcal{B}}(x), y)] \le \epsilon_{\mathcal{B}} + \mathbb{E}[\mathcal{L}(\hat{\phi}_{\mathcal{B}}, \psi_{\mathcal{B}}^*(x), y)] + 2R_{n_{\mathcal{B}}}(\mathcal{L} \circ \Psi \circ \hat{\phi}_{\mathcal{B}}) + 2L\sqrt{\frac{2\ln(2/\delta)}{n_{\mathcal{B}}}} \tag{88}$$

Take $1 - \delta/2$ for equation 88 and equation 86, we get with probability $1 - \delta$,

$$\mathbb{E}[\mathcal{L}(\hat{\psi}_{\mathcal{B}} \circ \hat{\phi}_{\mathcal{B}}(x), y)] \leq \kappa B \cdot d_{TV}(\mathbb{P}_{\phi_{\mathcal{B}}^*}, \mathbb{P}_{\phi_{\mathcal{A}}^*}) + \kappa \epsilon_{\mathcal{AB}} + \epsilon_{\mathcal{B}} \tag{89}$$

$$+ 2\kappa R_{m^2}(\mathcal{L}_{\text{CMD}} \circ \Phi_{\mathcal{B}}) + 2R_{n_{\mathcal{B}}}(\mathcal{L} \circ \Psi_{\mathcal{B}} \circ \hat{\phi}_{\mathcal{B}}) \tag{90}$$

$$+ 3\kappa B \cdot \sqrt{\frac{1}{n_{\mathcal{A}}^2} \ln \frac{4N_{[]}(\mathcal{P}_{\mathcal{X}_{\mathcal{A}} \times \mathcal{S}}(\Phi_{\mathcal{A}}), \frac{1}{n_{\mathcal{A}}^2})}{\delta}} + \kappa L \sqrt{\frac{2 \ln(4/\delta)}{m^2}} + 2L \sqrt{\frac{2 \ln(4/\delta)}{n_{\mathcal{B}}}} \tag{91}$$

$$\square$$

# B  DISCUSSION ABOUT CMC LOSS

In order to introduce a similar bound for the CMC loss, we introduce a likelihood bound assumption,

**Assumption B.1.** *For any fixed $\phi$, we have*

$$-\log \frac{p_{\phi_{\mathcal{A}}, \phi_{\mathcal{B}}}(\boldsymbol{x})}{p_{\hat{\phi}_{\mathcal{A}}, \phi_{\mathcal{B}}}(\boldsymbol{x})} \leq -(p_{\phi_{\mathcal{A}}}(\boldsymbol{x}) - p_{\hat{\phi}_{\mathcal{A}}}(\boldsymbol{x})) \log p_{\phi_{\mathcal{B}}}(\boldsymbol{x}) \tag{92}$$

*where*

$$p_{\phi_{\mathcal{A}}, \phi_{\mathcal{B}}}(\boldsymbol{x}) = \frac{\exp(z_i^{\mathcal{A}} \cdot z_j^{\mathcal{B}}/\tau)}{\sum_t \exp(z_t^{\mathcal{A}} \cdot z_j^{\mathcal{B}}/\tau)}, \quad p_{\phi_{\mathcal{A}}}(\boldsymbol{x}) = \frac{\exp(z_i^{\mathcal{A}} \cdot z_j^{\mathcal{A}}/\tau)}{\sum_t \exp(z_t^{\mathcal{A}} \cdot z_j^{\mathcal{A}}/\tau)} \tag{93}$$

From the proofs above, we can find that changing the CMD loss to CMC loss does not affect the lemmas and theorems other than the final results Theorem 3.3. Thus, we just show that with the assumption B.1 we can get the same result as the Theorem 3.3 with CMC loss.

*Proof.* Noticing that the main difference for CMD and CMC losses are between equation 94 and equation 105. We only discuss the bounded process here and other derivations should be the same.

$$\mathbb{E}[\mathcal{L}(\hat{\phi}_{\mathcal{B}}, \psi_{\mathcal{B}}^*(x), y)] \leq \kappa \mathbb{E}[\mathbb{E}_{x'}[\mathcal{L}_{\text{CMC}}(\hat{\phi}_{\mathcal{B}}, \phi_{\mathcal{B}}^*, (x, x'), s)]] \tag{94}$$

$$= \kappa \mathbb{E}[\mathcal{L}_{\text{CMC}}(\hat{\phi}_{\mathcal{B}}, \phi_{\mathcal{B}}^*, (x, x'), s)] \quad (\text{denote } \boldsymbol{x} = (x, x')) \tag{95}$$

$$= \kappa(\mathbb{E}[\mathcal{L}_{\text{CMC}}(\hat{\phi}_{\mathcal{B}}, \phi_{\mathcal{B}}^*, \boldsymbol{x}, s)] - \mathbb{E}[\mathcal{L}_{\text{CMC}}(\hat{\phi}_{\mathcal{B}}, \phi_{\mathcal{A}}^*, \boldsymbol{x}, s)] \tag{96}$$

$$+ \mathbb{E}[\mathcal{L}_{\text{CMC}}(\hat{\phi}_{\mathcal{B}}, \phi_{\mathcal{A}}^*, \boldsymbol{x}, s)] - \mathbb{E}[\mathcal{L}_{\text{CMC}}(\hat{\phi}_{\mathcal{B}}, \hat{\phi}_{\mathcal{A}}, \boldsymbol{x}, s)] \tag{97}$$

$$+ \mathbb{E}[\mathcal{L}_{\text{CMC}}(\hat{\phi}_{\mathcal{B}}, \hat{\phi}_{\mathcal{A}}, \boldsymbol{x}, s)]) \tag{98}$$

$$= \kappa(\mathbb{E}[-\log \frac{p_{\phi_{\mathcal{B}}^*, \hat{\phi}_{\mathcal{B}}}}{p_{\phi_{\mathcal{A}}^*, \hat{\phi}_{\mathcal{B}}}}(\boldsymbol{x}, s)] + \mathbb{E}[-\log \frac{p_{\phi_{\mathcal{A}}^*, \hat{\phi}_{\mathcal{B}}}}{p_{\hat{\phi}_{\mathcal{A}}, \hat{\phi}_{\mathcal{B}}}}(\boldsymbol{x}, s)] \tag{99}$$

$$+ \mathbb{E}[\mathcal{L}_{\text{CMC}}(\hat{\phi}_{\mathcal{B}}, \hat{\phi}_{\mathcal{A}}, \boldsymbol{x}, s)]) \tag{100}$$

$$\leq \kappa(\mathbb{E}[-p_{\phi_{\mathcal{B}}^*}(\boldsymbol{x}, s) \log p_{\hat{\phi}_{\mathcal{B}}}(\boldsymbol{x}, s) + p_{\phi_{\mathcal{A}}^*}(\boldsymbol{x}, s) \log p_{\hat{\phi}_{\mathcal{B}}}(\boldsymbol{x}, s)] \tag{101}$$

$$+ \mathbb{E}[-p_{\phi_{\mathcal{A}}^*}(\boldsymbol{x}, s) \log p_{\hat{\phi}_{\mathcal{B}}}(\boldsymbol{x}, s) + p_{\hat{\phi}_{\mathcal{A}}}(\boldsymbol{x}, s) \log p_{\hat{\phi}_{\mathcal{B}}}(\boldsymbol{x}, s)] \tag{102}$$

$$+ \mathbb{E}[\mathcal{L}_{\text{CMC}}(\hat{\phi}_{\mathcal{B}}, \hat{\phi}_{\mathcal{A}}, \boldsymbol{x}, s)]) \tag{103}$$

$$\leq \kappa B \cdot \left( d_{TV}(\mathbb{P}_{\phi_{\mathcal{B}}^*}, \mathbb{P}_{\phi_{\mathcal{A}}^*}) + d_{TV}(\mathbb{P}_{\hat{\phi}_{\mathcal{A}}}, \mathbb{P}_{\phi_{\mathcal{A}}^*}) \right) \tag{104}$$

$$+ \kappa \mathbb{E}[\mathcal{L}_{\text{CMC}}(\hat{\phi}_{\mathcal{B}}, \hat{\phi}_{\mathcal{A}}, \boldsymbol{x}, s)] \tag{105}$$

$$\tag{106}$$

Then we get the same convergence bound of CMC loss as the CMD loss as shown in Theorem 3.3

$$\square$$

**Further improvement**. The assumption B.1 in this paper is not trivial or prior to the analysis, further work to this research can focus on a better proof and result with CMC loss.

## C    DETAILED SETTINGS OF EXPERIMENTS

In this section, we give the detailed settings of datasets and training. In our experiments, all cross-modality distillation using a self-supervised learned ResNet on ImageNet. As mentioned in the paper, we only used the well-trained model provided by the official SimCLR with different structures of ResNet50, ResNet50(2x), and ResNet50(4x) but not using the ImageNet data in the distillation. To clarify the cross-modality distillation process, we give the dataset used for transferring and the detailed setting of the downstream task of each pair of modalities.

| Training Dataset | Sketchy | TUBerlin | Sketchy-Eval |
|---|---|---|---|
| Train/Test Split | 48,290 | 15,000/5,000 | 60,335/15,146 |
| Paired data $M$ | – | 15,000 | – |
| Optimizer | Adam | Adam | Adam |
| Optimizer Hyper-parameter | (0.9,0.999) | (0.9,0.999) | (0.9,0.999) |
| Learning Rate Schedule | None | Multi-Step(60,70,80) | Multi-Step(60,70,80) |
| Learning Rate | 1e-3 | 1e-3 | 1e-3 |
| Epoch | 100 | 100 | 100 |
| Batch Size | 64 | 64 | 64 |

Table 4: Details of image-sketch Distillation.

Since there are multiple sketches corresponding to one image in the Sketchy dataset, we consider all these pairs as positive pairs resulting in a total of 48290 training data. The train/split for TUBerlin and Sketchy-Eval just follows the typical setting used in Yu et al. (2017); Lin et al. (2020). Sketches in Sketchy-eval may have been trained without labels in distillation.

| Training Dataset | NYU-Depth V2 | NYU-Depth V2-Eval( Disjoint ) |
|---|---|---|
| Train/Test Split | 795 | 795/654 |
| Paired data $M$ | 795 | – |
| Optimizer | Adam | Adam |
| Optimizer Hyper-parameter | (0.9,0.999) | (0.9,0.999) |
| Learning Rate Schedule | None | Multi-Step(60,70,80) |
| Learning Rate | 1e-3 | 1e-2 |
| Epoch | 100 | 100 |
| Batch Size | 16 | 16 |

Table 5: Details of image-depth map Distillation.

For the image-depth map task, we only use the training data in NYU-Depth V2 and also use the labeled version in downstream segmentation.

| Training Dataset | VGGSound | VGGSound-Eval (Disjoint) |
|---|---|---|
| Train/Test Split | 4,625 | 10,000/10,000 |
| Paired data $M$ | 4,625 | – |
| Optimizer | Adam | Adam |
| Optimizer Hyper-parameter | (0.9,0.999) | (0.9,0.999) |
| Learning Rate Schedule | None | Multi-Step(60,70,80) |
| Learning Rate | 1e-3 | 1e-2 |
| Epoch | 100 | 100 |
| Batch Size | 16 | 16 |

Table 6: Details of video-audio Distillation.

In this case, we sample 4625 pairs of video and audio, translating the video into 12 frames and audio into spectrograms. A disjoint 10000 audio-only dataset is sampled to fine-tune downstream event classification where another 10000 are used for testing.

# D   MORE ABLATION STUDY AND DISCUSSION

## D.1   COMPARISON OF PERFORMANCE IMPROVEMENT AND ESTIMATED TOTAL VARIATION DISTANCE

| Tasks | image-sketch | | video-audio | image-depth map |
|---|---|---|---|---|
| | Sketchy | TUBerlin | VGGSound | NYU-Dpeth V2 |
| TV Distance | 0.04 | 0.04 | 0.10 | 0.06 |
| LE($\Delta$) | 8.30 | 11.06 | 9.82 | 3.48 |
| FT($\Delta$) | 1.73 | 3.38 | 2.46 | 3.47 |

Table 7: Comparison of Estimated TV distance. **LE** means linear evaluation; **FT** means fine-tuning.

To provide object evidence about our theory, we compute the estimated total variation distances and performance improvements of several datasets and tasks, specifically, we use the performance difference between CMD + LE and SSL + LE as LE($\Delta$), CMD + FT and Sup FT as FT($\Delta$). As shown in Table 7, our discussion in Section 4.1 actually fits with the estimated TV distance that image-sketch has a smaller modality gap than video-audio. The performance improvements of image-sketch are more significant than video-audio from the results in Table 7. Since the downstream task of image-depthmap datasets is segmentation which is different from the classification task as image-sketch and video-audio, it is unfair to compare the results directly with the other two cases. But in general, the quantitative results we provide here conform to the discussion in Section 4.1.

## D.2   INTERPOLATION OF CMC AND CMD LOSSES

| Datasets | Methods | $\alpha$CMC + (1-$\alpha$)CMD | | | | |
|---|---|---|---|---|---|---|
| | | $\alpha$=0 | $\alpha$=0.25 | $\alpha$=0.5 | $\alpha$=0.75 | $\alpha$=1 |
| Sketchy | LE | 72.61 | 72.97 | **73.45** | 73.19 | 73.24 |
| | FT | 85.63 | 85.32 | 86.91 | 87.34 | **87.54** |
| TUBerlin | LE | 65.70 | 67.04 | 67.54 | 67.62 | **68.72** |
| | FT | **77.86** | 77.36 | 77.18 | 77.62 | 77.44 |

Table 8: Comparison of combination of CMC and CMD losses. **LE** means linear evaluation; **FT** means fine-tuning.

We further conduct an ablation study to investigate how the combination affects the final performance. From Table 8, we can find that combining CMC and CMD losses may help the distillation on some tasks, but there is no dominant choice for every task. Theoretically, I think the combination will make the analysis much more difficult without any more assumptions. We think this is beyond the contribution of the work and can be investigated in future work.

## D.3 OUR METHOD WITH ViT MODELS

| Model | Method | image-sketch | | video-audio |
|---|---|---|---|---|
| | | Sketchy | TUBerlin | VGGSound |
| ViT-S/16 | SSL + LE | 63.20 | 42.14 | 15.97 |
| | CMC + LE | 68.11 | 47.76 | 18.94 |
| ViT-B/16 | SSL + LE | 66.19 | 44.38 | 16.33 |
| | CMC + LE | 70.83 | 50.32 | 22.79 |

Table 9: Comparison of ViT models. We use top-1 accuracy(%) for recognition tasks and mean IoU (%) for the segmentation task. **LE** means a linear evaluation on only the final classification layer; **FT** means fine-tuning the whole network.

We added experiments on ViT-S/16 and ViT-B/16 to show the effectiveness of our method. From the results in Table 9, we can find the base performances of ViT are lower than ResNet with a small number of training data while our method still helps the learning in the target modality with about 4-7% improvement.

## D.4 COMPARISON OF MOMENTUM DISTILLATION AND WITHIN-REGULARIZER

| Tasks | image-sketch | | video-audio | image-depth map |
|---|---|---|---|---|
| | Sketchy | TUBerlin | VGGSound | NYU-Dpeth V2 |
| M-CMC + LE | 73.21 | **68.80** | 28.33 | 17.98 |
| CMC + WR + LE | 72.61 | 68.58 | **28.47** | 16.75 |
| CMC + LE | **73.24** | 68.72 | 28.27 | **18.35** |
| M-CMC + FT | **87.73** | **77.76** | 35.27 | 27.68 |
| CMC + WR + FT | 87.19 | 76.98 | 34.78 | 26.27 |
| CMC + FT | 87.54 | 77.44 | **35.37** | **27.93** |

Table 10: Comparison of Momentum Distillation and Within-Regularizer. We use top-1 accuracy(%) for recognition tasks and mean IoU (%) for the segmentation task. **LE** means a linear evaluation on only the final classification layer; **FT** means fine-tuning the whole network. **M-CMC** means the CMC loss with momentum update, **WR** means within-regularizer in the target modality.

Inspired by the works Li et al. (2021); Lin et al. (2023), we test the momentum distillation and within-regularizer techniques with our method. M-CMC indicates the CMC loss combined with the momentum update. In Table 10, we show that there is not an obvious improvement when we use the momentum update in the distillation process. Specifically, when applying the momentum update to image-sketch modalities it helps the distillation to some extent but for the other two, there is no improvement. Furthermore, we try the within-regularizer in the target modality when distillation, denoted as WR. In some cases, the WR actually helps the distillation like VGGSound on LE, but gives no improvement or a similar result on other tasks. The experiment results just fit with the arguments we made above and show the difficulty of applying such techniques to our problem.

## D.5 COMPARISON WITH LIT

| Model | Method | image-sketch | | video-audio |
|---|---|---|---|---|
| | | Sketchy | TUBerlin | VGGSound |
| ViT-B/16 | SSL + LE | 66.19 | 44.38 | 16.33 |
| | CMC + LE | 70.83 | 50.32 | 22.79 |
| | LiT(Sup) | 73.19 | 56.68 | 22.15 |

Table 11: Comparison of LiT models. We use top-1 accuracy(%) for recognition tasks. **LE** means a linear evaluation on only the final classification layer; **FT** means fine-tuning the whole network.

We also conducted an experiment to compare our method and LiT. From Table 11, when we only test the image-sketch classification task LiT with a supervised image encoder actually has a better performance. This may caused by the classification labels of the images and sketches in the ImageNet and Sketchy/TUBerlin overlap. But in another way, it is not feasible to solve the segmentation task with the standard ViT framework since the feature can not be used to predict pixel-wise labels. The results show that LiT are more specific model rather than a generalizable framework.

