# OpenReview forum: "Generalizable Cross-Modality Distillation with Contrastive Learning"
_ICLR.cc/2024/Conference — Submitted to ICLR 2024_

### Official Review · Reviewer_QE51 · 2023-10-25

**Soundness:** 3 good
**Presentation:** 3 good
**Contribution:** 2 fair
**Rating:** 5
**Confidence:** 4

**Summary:**

This paper proposes a framework for cross-modality distillation, which aims to transfer knowledge from a source modality with rich information to a target modality with limited information. The framework leverages contrastive learning to exploit both positive and negative relationships in the paired data, and distills generalizable features for various downstream tasks. The paper also provides theoretical analysis and empirical results to support the effectiveness and versatility of the proposed method.

**Strengths:**

1. This paper is well-written and easy to follow.
2. The paper provides theoretical analysis and empirical results to support the effectiveness and versatility of the proposed method across diverse modalities (e.g., images, sketches, depth maps, videos, and audio) and tasks (e.g., recognition and segmentation).

**Weaknesses:**

1. While the paper reviews relevant literature on cross-modality distillation and contrastive learning, it omits references to recent works on multi-modal distillation, specifically [1-2], which employ an online distillation strategy different from the approach presented here. The authors are encouraged to provide a comprehensive comparison with these works. How does this paper compare with these works? Are there any advantages or disadvantages of using different distillation strategies and loss functions?  [1-2] appear closely related to this work, and it would be valuable to engage in a detailed discussion with these papers, especially regarding the learning of positive and negative relationships during distillation.
2. I appreciate the theoretical results of this paper but the two distillation losses, CMD and CMC, appear somewhat simplistic.. The CMD loss is just the cross entropy loss and the CMC loss is exactly the CLIP loss. Moreover, the CMD loss can be seen as a within-modal regularizer of multi-modal learning, which has been used in [2-3]. Consequently, it seems that these losses have been adopted from the multi-modal learning community with minimal modification, potentially diminishing the novelty and significance of the proposed cross-modality contrastive distillation framework. What is the novelty and significance of the proposed framework? How does it differ from existing methods for multi-modal learning or cross-modality transfer?
3. The paper neglects to discuss recent work such as LiT [4], which employs a locked image model in multi-modal tuning. The concept of the locked operator in LiT appears akin to cross-modality distillation in this paper. A detailed comparison between this paper and LiT, in terms of methodology and performance, is essential to elucidate the distinctions and commonalities between the two approaches.

**Questions:**

The main points to address in the rebuttal primarily stem from the "weaknesses" section we discussed earlier. Specifically, it would greatly benefit our understanding if the authors could provide a more extensive explanation of their method's contribution, ideally through a detailed comparison with the works below.

[1] Align before Fuse: Vision and Language Representation Learning with Momentum Distillation, NeurIPS 2021

[2] Graph Matching with Bi-level Noisy Correspondence, ICCV 2023

[3] CrossCLR: Cross-modal Contrastive Learning For Multi-modal Video Representations

[4] LiT: Zero-Shot Transfer with Locked-image text Tuning, CVPR 2022

---

> ### Author Response · Authors · 2023-11-18
> **Response (1/3)**
>
> We thank the reviewer for the thoughtful assessment and for the positive evaluation of our work. We address the points raised in the review below.
>
> **W1: A comprehensive comparison with recent works on multi-modal distillation**
>
> *Response:* Thanks a lot for the suggestion. We will explain the difference between our method and these two works and then elaborate on the difficulty of using these two methods in our problem. Furthermore, we conducted an experiment on the momentum distillation technique in both works to show that it does not give an obvious improvement in the performance but introduces a computational burden.
>
> The ALBEF[1] introduces a contrastive loss to align image and text features, which enables more grounded vision and language representation learning. Besides, they propose momentum distillation to improve learning from noisy web data. The alignment process in the ALBEF is to make the image and text features more grounded so that the features can be used for further cross-modal applications. However, in our problem, we want to distill the knowledge from a 'rich' modality to a 'limited' modality and aim at boosting the model performance only in the target 'limited' modality. Since the limited number of training data in the target modality, it may be difficult to construct a general feature space for both modalities. As the second contribution of ALBEF, the momentum distillation is designed to leverage a larger uncurated web dataset which is actually unavailable in our problem since the data in the target modality is limited. The key idea of our work is to fully investigate the relationship behind the paired data while the idea of momentum distillation is to eliminate the influence of noisy data which are not so compatible. In conclusion, the ALBEF is mainly designed for image-text modalities with large noisy paried data and its modules are not so feasible for the task we study.
>
> The COMMON[2] is developed for the graph matching problem which introduces quadratic contrastive loss consisting of traditional infonce loss, within-graph consistency loss, and cross-graph consistency loss. Another technique momentum distillation is also designed for robust matching which aims at eliminating the noisy label in a large dataset. In our problem, the number of available data is limited which can be only 700 in image-depthmap scenario in which case such a momentum update may not introduce obvious benefits in performance but additional computation cost.
>
> We added the experiment in the Appendix D.4 in the revised version. We test the momentum distillation and within-regularizer techniques with our method. M-CMC indicates the CMC loss combined with the momentum update. In Table 10, we show that there is not an obvious improvement when we use the momentum update in the distillation process. Specifically, when applying the momentum update to image-sketch modalities it helps the distillation to some extent but for the other two, there is no improvement. Furthermore, we try the within-regularizer in the target modality when distillation, denoted as WR. In some cases, the WR actually helps the distillation like VGGSound on LE, but gives no improvement or a similar result on other tasks. The experiment results just fit with the arguments we made above and show the difficulty of applying such techniques to our problem.

---

> > ### Author Response · Authors · 2023-11-18
> > **Response (2/3)**
> >
> > **W2: What is the novelty and significance of the proposed framework?**
> >
> > *Response:* Compared to the existing methods of multi-modal learning or cross-modality transfer, our framework has the following novelty and significance.
> >
> > (1). Our framework is a generalizable framework in several different ways. First, most existing frameworks of multi-modal only focus on two or more fixed modalities like image-text[1,3,4] or image-graph[2], many modifications are made to adapt to the specific modalities e.g., the momentum distillation is designed to leverage large web data of text and images. However, our framework can work to various modalities as the experiment results demonstrate. In another way, many cross-modality transfer works [5-8] rely on the supervised source modality to transfer the information to the target modality. Nevertheless, our framework can take advantage of self-supervised learning in the source modality to distill knowledge which makes the method feasible for more modalities. Finally, compared to the cross-modality transfer or multi-modal learning that is restricted to a specific task, e.g., classification, the feature space of the target modality that is distilled from the source modality of our method can adapted to diverse downstream tasks as our results show. In fact, when concentrating on some fixed modality or task, it will be easy to add modality-specific modification to our framework to gain performance improvement.
> >
> > (2). Effectiveness of using paired data and unlabeled data. Another difference between our framework and multi-modal/cross-modality learning is that our method is designed to distill information from a 'rich' modality to a 'limited' modality via only *a small number* of paired data, but works like [1-3] needs a large number of paired data to distill or learn general pattern space. Since the limited number of training data in the target modality, the method like within-modal regularizer [2-3] is not compatible with our method as we demonstrate in Q1.
> >
> > (3). Our method is theoretically bounded. To the best of our knowledge, this is the first work to establish a theoretical convergence analysis of such a contrastive learning-based distillation framework which is a key contribution of our work. Moreover, there are no theoretical results of the convergence or boundness of the previous methods such that we can not compare the theoretical performance of our method and previous methods.
> >
> > [5] Saurabh Gupta, Judy Hoffman, and Jitendra Malik. Cross modal distillation for supervision transfer. In Proceedings of the IEEE conference on computer vision and pattern recognition, pp. 2827–2836, 2016.
> > [6] Fida Mohammad Thoker and Juergen Gall. Cross-modal knowledge distillation for action recognition. In 2019 IEEE International Conference on Image Processing (ICIP), pp. 6–10. IEEE, 2019.
> > [7]Rui Wang, Zuxuan Wu, Zejia Weng, Jingjing Chen, Guo-Jun Qi, and Yu-Gang Jiang. Cross-domain contrastive learning for unsupervised domain adaptation. IEEE Transactions on Multimedia,2022.
> > [8]Zihui Xue, Sucheng Ren, Zhengqi Gao, and Hang Zhao. Multimodal knowledge expansion. In Proceedings of the IEEE/CVF International Conference on Computer Vision, pp. 854–863, 2021.

---

> > > ### Author Response · Authors · 2023-11-18
> > > **Response (3/3)**
> > >
> > > **W3: A detailed comparison between this paper and LiT**
> > >
> > > *Response*: Thanks for the suggestion. Though the fixed and cross-modality process is similar between our method and LiT, there are several key differences that indicate the novelty and contribution of our work.
> > >
> > > (1). The LiT is deployed to image-text modality only and the cross-distillation is conducted with a large number of image-text pairs which is designed to improve zero-shot transfer performance. However, our method is designed to improve the performance of a target modality with a small number of training data and paired data. From this view, the goals and the numbers of paired training data vary a lot between our method and LiT.
> > >
> > > (2). Our framework is a generalizable framework for different modalities and downstream tasks. The main experiment results of the LiT focus on the zero-shot classification and image-text modalities while in our work, we prove and demonstrate our method can work for diverse modalities and downstream tasks. In another way, the base LiT model uses a supervised pre-trained image encoder to increase the performance of zero-shot classification tasks while in our work we do not add such constraints and make the framework more generalizable.
> > >
> > > (3). Our method is theoretically bounded. To the best of our knowledge, this is the first work to establish a theoretical convergence analysis of such a contrastive learning-based distillation framework which is a key contribution of our work. However, there is no theoretical analysis of LiT models.
> > >
> > > We added the experiment in the Appendix D.5 in the revised version. We conducted an experiment to compare our method and LiT. From Table 11, when we only test the image-sketch classification task LiT with a supervised image encoder actually has a better performance. This may caused by the classification labels of the images and sketches in the ImageNet and Sketchy/TUBerlin overlap. But in another way, it is not feasible to solve the segmentation task with the standard ViT framework since the feature can not be used to predict pixel-wise labels. The results show that LiT are more specific model rather than a generalizable framework.
> > >
> > > **Q1:  A detailed comparison with the works below**
> > >
> > > *Response:* We have done a thorough comparison with the methods in the review in Response W1, W3, and Response W2 restates our novelty. Besides, we add more ablation study and discussion in the revised draft that may demonstrate the novelty of our method.

---

> > > > ### Author Response · Authors · 2023-11-21
> > > >
> > > > Thank you again for the time you took to review our work!
> > > >
> > > > We believe we have ave made comprehensive comparisons, both in terms of methodology and experiments with reference works like [1-4] and LiT. Experiment results can be found in Appendix D in the revised version. Hope the explanation can address your issues.
> > > >
> > > > We would really appreciate it if you could have a look at our replies and let us know if you have further questions/comments.
> > > >
> > > > We hope to be able to effectively utilize the discussion phase in case you have any further questions or comments about our work.
> > > >
> > > > Best regards.

---

> > > > ### Comment · Reviewer_QE51 · 2023-11-22
> > > >
> > > > I appreciate your response, and I find that some aspects of your reply have addressed my concerns. However, my primary concern remains the lack of novelty in the overall contribution. The claim that your "approach is applicable to various patterns" and “can utilize paired and unlabeled data” does not seem to constitute a particularly strong novelty. While you have presented a thorough theoretical analysis with rich bounds, I believe there is insufficient innovation in the proposed methodology, i.e., two losses. Therefore, I have decided to maintain my initial evaluation. I am open to considering feedback from other reviewers on this matter.

---

### Official Review · Reviewer_qhgx · 2023-10-29

**Soundness:** 3 good
**Presentation:** 3 good
**Contribution:** 2 fair
**Rating:** 5
**Confidence:** 3

**Summary:**

This paper proposes a cross-modality distillation method with contrastive learning. Existing self-supervised methods leverage a few pairwise unlabeled data to distill the knowledge by aligning features or statistics between the source and target modalities.  The Cross-Modality Contrastive Distillation (CMCD) framework proposed in this paper considers the unpaired and unlabeled data in source and target modalities. The convergence analysis reveals that the distance between source and target modalities significantly impacts the test error on downstream tasks within the target modality which is also validated by the empirical results.

**Strengths:**

1. This paper considers that mass multi-modality data is not paired in the real world. For memory and privacy-restricted scenarios where labeled training data is generally unavailable, this setting will be more practical. This motivation may stimulate follow-up research.

2. There is sufficient convergence analysis and detailed settings of experiments in the paper.

3. The experiments cover several modalities, such as image, sketch, depth map, and audio, and two downstream tasks of recognition and segmentation.

**Weaknesses:**

1. The proposed method follows the self-supervised knowledge distillation framework which is widely used in existing works. The common approach is using contrastive learning between source modality and target modality, and fine-tuning on labeled data in target modality, such as Ref[1].

2. The authors claim that both positive and negative correspondence are leveraged in the abstract. However, it seems that the negative correspondence is only used in the pretrain stage of source modality.

3. There are too few ablation experiments in the paper, e.g., it can be seen that different ResNet backbone networks cause less impact in Table 2.  I'm wondering if the transformer will make a big improvement.

 [1] Jin W, Lee D H, Zhu C, et al. Leveraging visual knowledge in language tasks: An empirical study on intermediate pre-training for cross-modal knowledge transfer[J]. arXiv preprint arXiv:2203.07519, 2022.

**Questions:**

Q1: Why not utilize unpaired data in the target modality?

Q2: If there is no labeled data in the target domain, is the model still effective?

Q3: In Table 1, what’s the downstream task of each dataset? It’s not clear to me in its current form.

---

> ### Author Response · Authors · 2023-11-18
>
> We thank the reviewer for the thoughtful assessment and for the positive evaluation of our work. We address the points raised in the review below.
>
> **W1: The proposed method follows the self-supervised knowledge distillation framework which is widely used in existing works**
>
> *Response:* Compared to the existing methods of multi-modal learning or cross-modality transfer, e.g., [1], our framework has the following novelty and significance.
>
> (1). Our framework stands out for its generalizability in multiple ways. Unlike the framework in [1], which focuses on fixed modalities like image-text and requires specific modifications for each modality, our framework demonstrates its ability to work with various modalities, as supported by our experimental results. Moreover, our method allows the distilled feature space from the source modality to be adapted to diverse downstream tasks, as evidenced by our results.
>
> (2). Effective utilization of paired and unlabeled data. Another notable distinction between our framework and [1] is the effective utilization of data. Our method is designed to distill information from a 'rich' modality to a 'limited' modality using only a small number of paired data, whereas [1] relies on a large number of paired data for distillation or learning general pattern space.
>
> (3). Theoretical foundation. Our method is supported by a rigorous theoretical convergence analysis, which is a key contribution of our work. To the best of our knowledge, no previous work has established theoretical results for the convergence or boundedness of contrastive learning-based distillation frameworks like ours.
>
> **W2: However, it seems that the negative correspondence is only used in the pretrain stage of source modality.**
>
> *Response:* Compared to the losses like the CMKD which only minimizes the feature distance between the positive paired data between source and target modalities. The cross-modality contrastive loss will minimize the distance between positive paired data and maximize the distance between the negative paired data. Specifically, in the equation
>
> $-\log \frac{\exp(z_i^{\mathcal{A}} \cdot z_i^{\mathcal{B}}/\tau)}{\sum_{t} \exp(z_t^{\mathcal{A}} \cdot z_i^{\mathcal{B}}/\tau)} $
>
> $z_i^{\mathcal{A}}$ and $z_i^{\mathcal{B}}$ are positive paired data and $z_i^{\mathcal{A}}$ and $z_j^{\mathcal{B}}$ ($j\neq i$) are negative paired data. When minimizing the loss, we will expect $-z_i^{\mathcal{A}} \cdot z_i^{\mathcal{B}}$ to be small and $-\sum_{j\neq i} \exp(z_i^{\mathcal{A}} \cdot z_j^{\mathcal{B}})$ to be large in which case will minimizing the cosine distance of positive data but maximizing the cosine distance of negative data. In this way, we can fully leverage the positive and negative relationships or alignment between source and target modalities.
>
> **W3: There are too few ablation experiments in the paper.**
>
> *Response:* Thanks for your suggestion. We add more ablation experiments and discussions in Appendix D in the revised draft. For the transformer structure, we added experiments on ViT-S/16 and ViT-B/16 to show the effectiveness of our method. From the results in Table 9, we can find the base performances of ViT are lower than ResNet with a small number of training data while our method still helps the learning in the target modality with about 4-7\% improvement.
>
> **Q1: Why not utilize unpaired data in the target modality?**
>
> *Response:* In the experiments of images and sketches, the paired data and the finetuned data are disjoint as described in the experiments setup in Section 4 and detailed experiments setting in Appendix. C. The main idea of our framework is that we can learn a generalizable representation from a well-studied modality and the contrastive distillation with paired data. So based on the learned feature, we can still use the unpaired data in the target modality.
>
> **Q2: If there is no labeled data in the target domain, is the model still effective?**
>
> *Response:* Actually, our framework is primarily designed to learn a better and more generalizable representation/feature space in the target modality by using cross-modality contrastive distillation. And in this case, it does not need labeled data in the target domain. In experiments, we demonstrate this point by only training a linear layer upon the learned feature extractor and decreasing the number of labeled data as described in the last ablation study in Section 4.2. When only using 1-shot/5-shot labeled data in the target domain our method can achieve a similar performance with the supervised finetune.
>
> **Q3: In Table 1, what’s the downstream task of each dataset? It’s not clear to me in its current form.**
>
> *Response:* Thanks for the suggestion, we have described the downstream tasks in the Experiments Setup at the beginning of Section 4 Experiments and Discussion. To make it more clear, we have added the downstream tasks information in the Table 1 in the revised version.

---

> > ### Author Response · Authors · 2023-11-21
> >
> > Thank you again for the time you took to review our work!
> >
> > We believe that we have elaborated on the novelty of our method and explained how the positive and negative relationships are utilized in our framework. Furthermore, we provide more ablation experiments in Appendix D in the revised version including distillation with transformers, interpolation of losses, and comparison with other distillation techniques e.g., momentum update.
> >
> > We would really appreciate it if you could have a look at our replies and let us know if you have further questions/comments.
> >
> > We hope to be able to effectively utilize the discussion phase in case you have any further questions or comments about our work.
> >
> > Best regards.

---

> > > ### Comment · Reviewer_qhgx · 2023-11-23
> > > **Reply to the Authors's Responses**
> > >
> > > Thanks to the answers by the authors. After carefully reading your responses and the discussions with other reviewers, I would like to keep my original rating. Although I appreciated the theoretical guarantee of the paper, my major concern related to its novelty and distinctness from existing approaches remains.

---

### Official Review · Reviewer_3kmi · 2023-10-31

**Soundness:** 3 good
**Presentation:** 2 fair
**Contribution:** 3 good
**Rating:** 5
**Confidence:** 3

**Summary:**

This paper discusses cross-modality distillation for data with limited information. Existing methods focus on aligning features between source and target modalities but overlook negative relationships in unpaired data. The authors introduce "generalizable cross-modality contrastive distillation (CMCD)" that leverages both positive and negative correspondences, outperforming existing methods across various modalities and tasks. They emphasize the impact of modality distance on downstream task performance.

**Strengths:**

1. The proposed CMD and CMC losses seem to be novel.
2. Experimental results show the superiority of the proposed two losses over previous methods.
3. Theoretical analysis shows that the performance of proposed two losses are controlled by the alignment of latent feature distributions.

**Weaknesses:**

1. The motivation for the proposed distillation is not clear to me. If we do not use the labels of the source modality, only do self-supervised learning on the source data (though source and target data are paired, in my understanding they share labels), and finally only supervised trained on target data, why do we expect improvements (though results are improved)? The pipeline of SSL (source), alignment (source + target), and FT (target) is not reasonable as we actually can direct SSL (target) + FT (target).
2. Theoretical analysis cannot support why the losses are useful as it only proves that the test error of the target task is bounded by the distance between two distributions. However, there is no theorem that the proposed losses can achieve a smaller distance than the previous method.
3. The experiments actually show that SSL (source) - alignment (source + target) - FT (target) is better than SSL (target) - FT (target) and FT (target), which is counterintuitive. I would like to see a more detailed analysis of why this happens.
4. Experimental details are missing, e.g., epochs/lr of SSL, alignment, and FT, which is essential for evaluating the results without reproducing the experiments.

Overall, too many details of the experiments and settings are missing. I would like to request authors add details for reproducibility.

**Questions:**

See weakness.

---

> ### Author Response · Authors · 2023-11-18
>
> We thank the reviewer for the thoughtful assessment and for the positive evaluation of our work. We address the points raised in the review below.
>
> **W1: The motivation for the proposed distillation is not clear**
>
> *Response*: The motivation or reason why our pipeline can work is that the number of data in the source modality can be much larger than the target modality which will lead to 'rich' knowledge in the source modality and 'limited' knowledge in the target modality as we mentioned in the introduction. From this view, the pipeline of SSL(source) + alignment(source + target ) can distill the 'rich' information from the SSL(source) via the alignment to the target modality which will be better to only leverage the 'limited' knowledge in the target modality, i.e., SSL(target) + FT(target). Specifically, in our experiments, we pre-trained the SSL model on ImageNet with more than 1 million images but only 60k sketches available in the target modality.
>
> **W2:Theoretical analysis cannot support why the losses are useful**
>
> *Response*: To the best of our knowledge, this is the first work to establish a theoretical convergence analysis of such a contrastive learning-based distillation framework which is a key contribution of our work. Moreover, there are no theoretical results of the convergence or boundness of the previous methods such that we can not compare the theoretical performance of our method and previous methods.
>
> **W3: I would like to see a more detailed analysis of why this happens**
>
> *Response*: We have explained this problem in the first response and want to provide another analysis of our method and why our method works. In simple words, SSL (source) - alignment (source + target) will provide a better pre-trained start point than SSL (target) - FT (target) or FT(target) since we leverage the much larger number of training data in source modality than the target modality.
> If we assume that good performance on the target(e.g., high-quality sketch) downstream tasks, e.g., classification is closely related to the goodness of the representation we learn in the target modality, then given a small number of training data in the target modality, it will be difficult to learn a comprehensive representation. However, there may exist a source modality (e.g., image w.r.t high-quality sketch) with a large number of available data. By leveraging enough training data and SSL in the source modality, we can get a well-learned and powerful representation space of the source modality. Alignment in our work will distill the information from the learned representation space to the target modality and help to learn a better representation space in the target modality. In this way, we will get a better pre-trained model for the tasks in the target modality.
>
> **W4: Experimental details are missing**
>
> *Response*: In fact, we have provided information like epochs/lr of SSL, alignment, and FT in Appendix.C on page 19. Besides, we also provide the code for reproducibility.

---

> > ### Author Response · Authors · 2023-11-21
> >
> > Thank you again for the time you took to review our work!
> >
> > We believe that we have elaborated on how our framework works with the distillation from the 'rich' modality to the 'limited' modality and the experimental details are provided in Appendix C as expected.
> >
> > We would really appreciate it if you could have a look at our replies and let us know if you have further questions/comments.
> >
> > We hope to be able to effectively utilize the discussion phase in case you have any further questions or comments about our work.
> >
> > Best regards.

---

### Official Review · Reviewer_s3Ct · 2023-11-05

**Soundness:** 3 good
**Presentation:** 3 good
**Contribution:** 3 good
**Rating:** 6
**Confidence:** 3

**Summary:**

This paper proposes a new method for a cross modality distillation problem. The proposed method is based on contrastive learning in order to take both positive instance pairs and negative pairs into account, while existing methods typically rely on the positive pairs. Furthermore, the paper provides theoretical analysis on the error bound of the proposed method. The effectiveness of the proposed method is verified on a wide variety of cross modal transfer setting.

**Strengths:**

- S1. The proposed approach is simple and reasonable, and it turns out to be effective in different cross modal transfer learning scenario.
- S2. The paper provides theoretical analysis on the error bounds and discusses the characteristic of the method based on the analysis, i.e., in which situations the proposed method is expected to work well.
- S3. The paper generally reads well.

**Weaknesses:**

- W1. The originality of the proposed method is not that outstanding because equation (2) is a straightforward adaptation of self-supervised distillation [Fang et al., 2021] to the cross modal setting and equation (3) is also a straightforward adaptation of what was proposed in the paper of CLIP [Radford et al., 2021]. I do acknowledge the theoretical analysis part, but the novelty of the method itself is rather limited.
- W2. The claim in the 2nd last line of section 3
> “It indicates that if source and target modalities have more common information or patterns,
the algorithm will have a higher probability of distilling more information from the source modality to the downstream task in the target modality."

    lacks objective evidence. The paragraph “Relationship with the generalization bound” in section 4.1 discusses it, but the discussion is rather subjective. It would become much more convincing if the authors can provide more objective evidence. For example, it may be interesting to provide the analysis on the relationship between the performance and estimated total variation distance between two datasets.
- W3. Some important experimental setting is not described in the main paper. What are the values of M and m?

Typo and minor suggestions.
1. In Figure 1, it is better to clearly indicates which figure corresponds to which method.
1. Please check the grammar of the sentence after equation (13)
> “Detailed proof our the Theorem 3.3 in Appendix A.3.”
1. The first sentence of section 4
>“To demonstrate the efficiency of our algorithm, we conduct extensive experiments on various cross-modality tasks.”

    efficiency -> effectiveness?
1. In P7, 3 lines from the bottom,
>“our method utilizing CMD/CMC loss achieves top-1 accuracies of 72.61%/73.24% on Sketchy, outperforming the best baseline by a margin of 3%.”

    The margin is less than 2% as SOCKET+LE achieves 71.33%.

**Questions:**

Is it possible to apply both CMD and CMC?

---

> ### Author Response · Authors · 2023-11-18
>
> We thank the reviewer for the thoughtful assessment and for the positive evaluation of our work. We address the points raised in the review below.
>
> **W1:The novelty of the method itself is rather limited**
>
> *Response*: Though the losses used for the cross-modality distillation are an adaption of self-supervised distillation and CLIP, our method is designed for a totally different task which is a provable and more generalizable framework. One of the key differences between our framework and self-supervised distillation/CLIP is that our method is designed to distill information from a 'rich' modality to a 'limited' modality via only *a small number* of paired data, but both self-supervised distillation and CLIP needs a large number of paired data to distill or learn general pattern space. In another way, self-supervised distillation only works for single-modality situations and CLIP is mainly designed for image language while our framework can work for any modalities only when there is some paired relationship that can be investigated. Besides, we give the theoretical analysis to further understand the algorithm which is also a key contribution of the work.
>
> **W2:The analysis on the relationship between the performance and estimated total variation distance between two datasets.**
>
> *Response*: Thanks for the suggestion. We add new experiments and discussions in Appendix D.1 in the revised draft. To provide object evidence, we compute the estimated total variation distances and performance improvements of several datasets and tasks, specifically, we use the performance difference between CMD + LE and SSL + LE as LE($\Delta$), CMD + FT and Sup FT as FT($\Delta$). As shown in Table 7, our discussion in Section 4.1 actually fits with the estimated TV distance that image-sketch has a smaller modality gap than video-audio. The performance improvements of image-sketch are more significant than video-audio from the results in Table 7. Since the downstream task of image-depthmap datasets is segmentation which is different from the classification task as image-sketch and video-audio, it is unfair to compare the results directly with other two cases. But in general, the quantitative results we provide here conform to the discussion in Section 4.1.
>
> **W3:What are the values of M and m?**
>
> *Response*: The values of $M$ and $m$ mean the number of the whole paired data and the number used in the distillation respectively. For example, in the ablation study, we may have a whole number of $M=15000$ paired image and sketch while we only use $m=3000$ in the distillation and then the $m/M=20\%$ means that we use 20\% data for distillation. We have added the explanation of $M$ and $m$ in the discussion and given the true values in detailed experiment settings in the appendix.
>
> **W4:Typos**
> Thanks a lot for the suggestion, we have fixed all typos in the updated version.
>
> **Q1:Is it possible to apply both CMD and CMC?**
>
> We add new experiments and discussions in Appendix D.2 in the revised draft. Absolutely, we can combine these two losses when distilling, we further conduct an ablation study to investigate how the combination affects the final performance. From Table 8, we can find that combining CMC and CMD losses may help the distillation on some tasks, but there is no dominant choice for every task. Theoretically, I think the combination will make the analysis much more difficult without any more assumptions. We think this is beyond the contribution of the work and can be investigated in future work.

---

> > ### Author Response · Authors · 2023-11-21
> >
> > Thank you again for the time you took to review our work!
> >
> > We believe that we have elaborated on the novelty of our work and added more ablation study results according to Weakness 2 and Question 1.
> >
> > We would really appreciate it if you could have a look at our replies and let us know if you have further questions/comments.
> >
> > We hope to be able to effectively utilize the discussion phase in case you have any further questions or comments about our work.
> >
> > Best regards.

---

> > > ### Comment · Reviewer_s3Ct · 2023-11-23
> > >
> > > I appreciate the authors' response.
> > > I have gone through the response as well as the discussion with the other reviewers.
> > > I basically agree with the other reviewers on the concern about the novelty of the paper as I also raised in W1, but I am still slightly leaning to acceptance because of the strength I raised as well as the additional results the authors provided.

---

### Author Response · Authors · 2023-11-18
**General Response to All Reviewers**

We would like to thank the reviewers for the positive feedback and suggestions on our work. Based on the reviews, we have added more ablation experiments to Appendix D in the revised draft(Titles in Blue). We hope that these results can further highlight the contributions of our paper.
Besides, We would like to clarify our contributions in this regard.

Compared to the existing methods of multi-modal learning or cross-modality transfer, our framework has the following novelty and significance.

(1). Our framework is a generalizable framework in several different ways. First, most existing frameworks of multi-modal only focus on two or more fixed modalities like image-text[1,3,4] or image-graph[2], many modifications are made to adapt to the specific modalities e.g., the momentum distillation is designed to leverage large web data of text and images. However, our framework can work to various modalities as the experiment results demonstrate. In another way, many cross-modality transfer works [5-8] rely on the supervised source modality to transfer the information to the target modality. Nevertheless, our framework can take advantage of self-supervised learning in the source modality to distill knowledge which makes the method feasible for more modalities. Finally, compared to the cross-modality transfer or multi-modal learning that is restricted to a specific task, e.g., classification, the feature space of the target modality that is distilled from the source modality of our method can adapted to diverse downstream tasks as our results show. In fact, when concentrating on some fixed modality or task, it will be easy to add modality-specific modification to our framework to gain performance improvement.

(2). Effectiveness of using paired data and unlabeled data. Another difference between our framework and multi-modal/cross-modality learning is that our method is designed to distill information from a 'rich' modality to a 'limited' modality via only *a small number* of paired data, but works like [1-3] needs a large number of paired data to distill or learn general pattern space. Since the limited number of training data in the target modality, the method like within-modal regularizer [2-3] is not compatible with our method as we demonstrate in Q1.

(3). Our method is theoretically bounded. To the best of our knowledge, this is the first work to establish a theoretical convergence analysis of such a contrastive learning-based distillation framework which is a key contribution of our work. Moreover, there are no theoretical results of the convergence or boundness of the previous methods such that we can not compare the theoretical performance of our method and previous methods.

[1] Align before Fuse: Vision and Language Representation Learning with Momentum Distillation, NeurIPS 2021

[2] Graph Matching with Bi-level Noisy Correspondence, ICCV 2023

[3] CrossCLR: Cross-modal Contrastive Learning For Multi-modal Video Representations

[4] LiT: Zero-Shot Transfer with Locked-image text Tuning, CVPR 2022

[5] Saurabh Gupta, Judy Hoffman, and Jitendra Malik. Cross modal distillation for supervision transfer, CVPR 2016.

[6] Fida Mohammad Thoker and Juergen Gall. Cross-modal knowledge distillation for action recognition, ICIP 2019.

[7] Rui Wang, Zuxuan Wu, Zejia Weng, Jingjing Chen, Guo-Jun Qi, and Yu-Gang Jiang. Cross-domain contrastive learning for unsupervised domain adaptation. IEEE Transactions on Multimedia,2022.

[8] Zihui Xue, Sucheng Ren, Zhengqi Gao, and Hang Zhao. Multimodal knowledge expansion, ICCV 2021.

---

### Author Response · Authors · 2023-11-21

Dear reviewers,
We again thank you for your time and valuable feedback.
Having just two days left for the discussion period, we would love to address any issues remaining following our author responses.

Best Wishes

---

### Meta-Review · Area_Chair_Z7Jb · 2023-12-12

**Metareview:**

The paper presents a new approach to cross-modality distillation, utilizing contrastive learning and providing a theoretical analysis of error bounds. While the reviewers appreciate the extensive experiments and clear presentation, they express concerns regarding the novelty of the methodology and the practical implications of the findings. The adaptations of existing techniques seem to offer limited advancement over current methods. Additionally, there is a lack of compelling evidence to support some of the paper's claims, and crucial experimental details for reproducibility are missing. The theoretical contributions, although noteworthy, do not sufficiently align with the practical applications to fully demonstrate the paper's impact. In essence, the paper revisits and refines existing concepts but falls short in delivering a compelling narrative or justification for its approach. It leaves the reader questioning the practical necessity and distinctiveness of the proposed framework in the broader context of current research.

**Justification For Why Not Higher Score:**

Limitations pointed out by the reviewers

**Justification For Why Not Lower Score:**

N/A

---

### Decision · Program_Chairs · 2024-01-16

Reject